# EEG SEIZURE DETECTION AND TRAFFIC FORECASTING WITH SPACE-TIME SELF-ATTENTION

## ABSTRACT

This work introduces a Transformer-based approach for graph signal processing that leverages a novel task-specific attention mechanism, namely *NT* Attention. Unlike conventional self-attention mechanisms, our method attends to all nodes across multiple time steps, enabling the model to effectively capture dependencies between nodes over extended time periods. This addresses a key limitation faced by traditional methods. Additionally, we propose geometry-aware masking (GMask), which incorporates the graph topology into the sparsification of the self-attention matrix. This enhances efficiency while preserving the rich temporal information conveyed by the nodes. We demonstrate the effectiveness of our approach on two critical applications: EEG seizure detection and traffic forecasting. Both tasks involve data collected from fixed sensors, such as electrodes or road sensors, where data from one sensor can influence others temporally and spatially. Our model enhances sensitivity in fast seizure detection by 20 percentage points compared to state-of-the-art and significantly outperforms current methods in traffic forecasting.

## 1 INTRODUCTION

A significant portion of the time series data we utilize in various machine learning applications is gathered thanks to fixed located sensors (Tang et al., 2021; Li et al., 2018). These sensors play a crucial role in collecting information for different applications such as healthcare and time-series forecasting (Jasper, 1958; Li et al., 2018). One notable application of these sensors is in neural recordings, such as electroencephalography (EEG) signals, where electrodes are placed on a patient's scalp (Jasper, 1958). Similarly, the deployment of traffic sensors along roadways for monitoring traffic flow is another significant example, greatly impacting our daily routines (Shao et al., 2022).

While these sensors remain fixed in their respective locations, it is crucial to recognize that the data flow from one sensor can influence others *both temporally and spatially* (Tang et al., 2021; Li et al., 2018). For example, effective EEG seizure detection require learning both $(i)$ long-range spatial dependencies, as seizure activity may originate at a focal electrode and then spread to other brain regions, and $(ii)$ long-range temporal dependencies, as if a seizure occurs at the beginning of the window with no subsequent activity, the model need to classify based on this brief episode. In the context of traffic data, successful traffic forecasting needs to capture both $(i)$ long-range spatial dependencies, as congestion at a major nodes can create ripple effects, and $(ii)$ long-range temporal dependencies, where forecasting typically operates with a 5-minute window resolution and forecasting 1 or 2 hours ahead involves processing 12 or 24 data windows, which is considered long-range for this task (Li et al., 2018; Yu et al., 2018; Shang et al., 2021; Zheng et al., 2020).

Motivated by these observations, data recorded from various fixed sensors or electrodes is often framed as a temporal graph representation, where the topology remains fixed over time. Several studies have leveraged different combinations of Graph Neural Network (GNN) and Recurrent Neural Network (RNN) to capture the spatio-temporal dynamics inherent in such data (Yu et al., 2018; zot; Song et al., 2020; Tang et al., 2021; Li et al., 2022; Ho & Armanfard, 2023). Two crucial applications, EEG-based seizure detection and traffic prediction, have received considerable attention in prior research (Tang et al., 2021; Ho & Armanfard, 2023; Li et al., 2018; Song et al., 2020). In this study, we focus on these applications due to their significant impact and relevance. Over 50 million worldwide suffer from epilepsy (WHO; Begley et al., 2022), highlighting the critical need for effective seizure detection and prevention methods (Shoaran et al., 2016). Traffic congestion

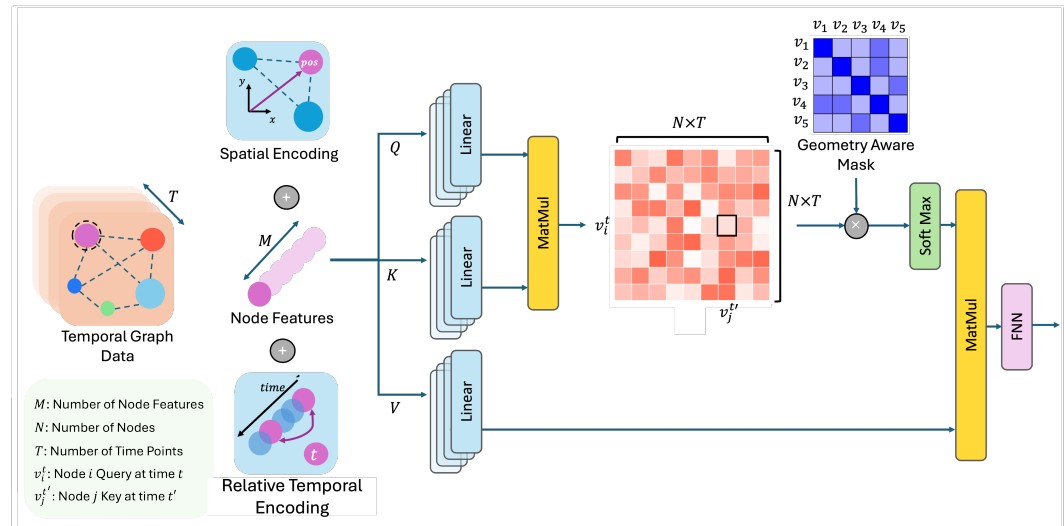

Figure 1: An illustration of our proposed temporal and spatial encoding, space-time self attention, and geometry-aware mask in *NT*Attention.

significantly impacts daily life, emphasizing the critical need for accurate forecasting to enhance transportation efficiency (Song et al., 2020).

While previous studies have modeled the spatio-temporal dynamics of temporal graphs using GNNs and RNNs (Li et al., 2018; Yu et al., 2018; Tang et al., 2021; Ho & Armanfard, 2023; Shao et al., 2022), they often struggle to account for long-range dependencies among distant nodes over extended periods, affecting their accuracy in capturing long-range space-time dependencies (Tang et al., 2021; Ho & Armanfard, 2023; Song et al., 2020). Attempts have been made to use attention for graphs, e.g., Graphormer (Ying et al., 2021) or Graph Attention Networks (GAT) (Velickovic et al., 2017), but they have not been investigated for graph signal representation with fixed nodes. While some approaches, such as (Guo et al., 2019), have integrated transformers to capture long-range temporal dependencies, they solely rely on node embeddings and graph convolutions for graph representation (Morris et al., 2019). This reliance can lead to decreased accuracy, especially for graphs with a large number of nodes, as it may struggle to effectively detect long-range space-time dependencies.

In this study, we propose a new, *simple* yet highly *effective* self-attention module, namely *NT*Attention, which attends to all graph nodes over extended time periods for learning spatio-temporal dynamics in temporal graphs. Furthermore, to enhance efficiency in the spatial domain, we propose a geometry-aware attention mask that ensures spatially distant nodes do not attend to each other. We illustrate our method in Fig. 1. We evaluate our model on two key applications in graph signal processing: EEG-based seizure detection and traffic forecasting. Our main contributions are as follows:

- We introduce a new self-attention mechanism, termed *NT*Attention, for graph signal processing, which effectively attends to all nodes across multiple time steps. Our Transformer design is easy to implement and *strongly motivated* by the characteristics of temporal graph signals. Our architecture excels at addressing long-range space-time dynamics, a challenge that previous methods in graph signal processing have struggled to overcome.

- We introduce Geometry-Aware Masking (GMask) for *NT*Attention, which *significantly enhances the computational efficiency* of our proposed attention mechanism, while simultaneously improving the model's generalization performance.

- Our model sets a new standard in both seizure detection and traffic forecasting. Specifically, it improves sensitivity for long-term seizure detection by 20 percentage points compared to previous methods, which is crucial for ensuring that seizure events are not missed. Additionally, it achieves an impressive MAE of 2.94 for 1-hour traffic forecasting, outperforming all other benchmarks while maintaining *similar memory* and *computational requirements*.

## 2 RELATED WORK

We concentrate on two key applications of graph signal processing: **1)** EEG-based seizure analysis, and **2)** Traffic forecasting. Accordingly, this section is divided into two parts to comprehensively review the related work and advancements of various baseline methods in each area.

**EEG-based Seizure Detection** Various studies have attempted to develop machine learning and deep learning models for EEG-based seizure detection (Asif et al., 2020; O'Shea et al., 2020; Ho & Armanfard, 2023; Shoaran et al., 2016; Tang et al., 2021; Yan et al., 2022). For instance, (Asif et al., 2020; Saab et al., 2020) used CNN-based architectures, either utilizing spectral features or treating EEG data as multi-channel images, which neglects the time-series structure of EEG signals. Furthermore, (Ahmedt-Aristizabal et al., 2020) employed a CNN-LSTM architecture that captures both spatial and temporal dependencies in EEG signals. However, these approaches overlook the non-Euclidean geometry inherent in EEG signals (Tang et al., 2021; Ho & Armanfard, 2023).

To address this, different variations of GNNs have been applied to the seizure detection task. For example, (Tang et al., 2021) employed two versions of the diffusion convolution recurrent neural network (DCRNN): one using a distance-based graph and the other a correlation-based graph. These approaches leverage GNNs to capture spatial information considering non-Euclidean geometry and RNNs for temporal dependencies. However, these models still face challenges in capturing long-range node and time dependencies within the EEG structure (Yan et al., 2022; Li et al., 2022). This limitation arises because RNNs and GNNs are adept at capturing local spatial and temporal information but struggle with long-range space-time dependencies (Vaswani et al., 2017). A detailed comparison of different models, highlighting their strengths and weaknesses, is provided in Table 1 and dataset description provided in Table 3.

**Traffic Forecasting** Traditional methods such as support vector regression (SVR) (Hong, 2011) and Vector Auto-regressive (VAR) (Ermagun & Levinson, 2018) have been utilized for traffic forecasting. These models, however, do not capture the information flow between different sensors and predict the traffic flow for each sensor solely based on the data available for that sensor (Tang et al., 2021). Deep fully connected neural networks have been employed for traffic forecasting tasks (Zhang et al., 2016), leveraging data from all sensors to predict traffic flow. However, these models do not account for the locality of sensor placement, which is crucial for capturing the spatial dependencies and variations in traffic patterns. Recurrent neural network (RNN) based models, such as Long Short-Term Memory (LSTM), have also been utilized for capturing the temporal information of traffic data but similarly overlook the locality of sensors (Hochreiter & Schmidhuber, 1997; Laptev et al., 2017). To better capture spatial information, convolutional neural network (CNN) architectures have been used to model the information flow between traffic sensors (Huang et al., 2022; Zhang et al., 2017).

Recent research has shown a growing interest in leveraging graph neural networks (GNNs) to address traffic forecasting challenges (Tang et al., 2021; Yu et al., 2018; Song et al., 2020; Zheng et al., 2020; Shang et al., 2021). However, existing GNN-based models struggle to effectively capture long space-time dependencies (Song et al., 2020; Wu et al., 2020). Inspired by developments in language models, some studies have integrated attention mechanisms with convolutional layers for this task (Guo et al., 2019), or introduced separate attention mechanisms for spatial and temporal representation (Zheng et al., 2020). Despite these advancements, existing models still exhibit low accuracy, particularly with long window sizes, and fail to incorporate long space-time correlations, such as how distant nodes affect each other over time. In Table 2, we summarize the advantages and disadvantages of current traffic forecasting models and in Table 4 we presented dataset description. Additionally, since the connection and similarity between seizure detection and traffic forecasting tasks have been established in Tang et al. (2021), but most models primarily focus on a single application, *NT*Attention has been specifically designed as a unified model capable of effectively handling both tasks.

## 3 *NT*ATTENTION

In this section, we present *NT*Attention, a new model for graph signal processing tasks. We begin in Section 3.1 by formalizing the problem setting and establishing the notations of signal processing on graphs. In Section 3.2, we enhance each node's features with spatial and temporal encoding based on

Table 1: **Comparison among seizure detection models: A)** Capturing Non-Euclidean geometry of EEG **B)** Capturing temporal nature of EEG **C)** Capturing long range time dependency **D)** Capturing long range electrode dependency **E)** Capturing long range electrode-time dependency

| Method | A | B | C | D | E |
|---|---|---|---|---|---|
| SeizureNet (Asif et al., 2020) | ✗ | ✓ | ✗ | ✗ | ✗ |
| LSTM (Hochreiter & Schmidhuber, 1997) | ✗ | ✓ | ✗ | ✗ | ✗ |
| Dense-CNN (Saab et al., 2020) | ✗ | ✗ | ✗ | ✗ | ✗ |
| CNN-LSTM (Ahmedt-Aristizabal et al., 2020) | ✗ | ✓ | ✗ | ✗ | ✗ |
| DCRNN (Tang et al., 2021) | ✓ | ✓ | ✓ | ✓ | ✗ |
| Transformer (Vaswani et al., 2017) | ✓ | ✓ | ✓ | ✗ | ✗ |
| REST (Afzal et al., 2024) | ✓ | ✓ | ✗ | ✗ | ✗ |
| *NT*Attention | ✓ | ✓ | ✓ | ✓ | ✓ |

Table 2: **Comparison among traffic forecasting models: A)** Capturing spatial dependency of traffic data **B)** Considering the graph geometry **C)** Capturing long range time dependency **D)** Capturing long range node dependency **E)** Capturing long range space-time dependency

| Method | A | B | C | D | E |
|---|---|---|---|---|---|
| HA | ✗ | ✗ | ✗ | ✗ | ✗ |
| VAR | ✗ | ✗ | ✗ | ✗ | ✗ |
| SVR | ✗ | ✗ | ✗ | ✗ | ✗ |
| FNN | ✓ | ✗ | ✗ | ✗ | ✗ |
| LSTM (Hochreiter & Schmidhuber, 1997) | ✓ | ✗ | ✗ | ✗ | ✗ |
| STGCN (Yu et al., 2018) | ✓ | ✓ | ✗ | ✗ | ✗ |
| DCRNN (Li et al., 2018) | ✓ | ✓ | ✗ | ✗ | ✗ |
| GTS (Shang et al., 2021) | ✓ | ✓ | ✓ | ✗ | ✗ |
| ASTGN (Guo et al., 2019) | ✓ | ✓ | ✓ | ✗ | ✗ |
| GMAN (Zheng et al., 2020) | ✓ | ✓ | ✓ | ✓ | ✗ |
| STAEFormer (Liu et al., 2023) | ✓ | ✓ | ✓ | ✓ | ✗ |
| PM-MemNET (Lee et al., 2021) | ✓ | ✓ | ✓ | ✓ | ✗ |
| *NT*Attention | ✓ | ✓ | ✓ | ✓ | ✓ |

Table 3: TUSZ data description

| Data split | EEG-Files (% Seizures) | Patients (% Patents with Seizures) |
|---|---|---|
| Training | 4664 (5.34%) | 579(36%) |
| Evaluation | 881(5.82%) | 43(79%) |
| Total | 5545 (5.41%) | 622 (39%) |

Table 4: METR-LA data description

| Data split | Samples | # Node | Time Span |
|---|---|---|---|
| Training | 23990 | 207 | 2.8 month |
| Evaluation | 6855 | 207 | 0.4 month |
| Testing | 3427 | 207 | 0.8 month |

its position in the graph and the relative time point at which data was collected. Then, in Section 3.3 we introduce our new attention mechanism tailored for temporal graphs. Finally, in Section 3.4 we introduce the geometry-aware masking of the attention matrix to enhance model performance and efficiency.

## 3.1 PROBLEM SETTING AND FORMULATION

We represent the sensor network as a graph $\mathcal{G} = \{\mathcal{V}, \mathcal{E}, \mathcal{P}\}$ with $\mathcal{V} = \{v_1, ..., v_N\}$ as the nodes, $\mathcal{E}$ representing the edges, and $\mathcal{P} = \{p_1, \ldots, p_n\}$ being a set of vectors $p_i \in \mathbb{R}^2$ representing node coordinates in space. We denote the data observed at time point $t$ on graph $\mathcal{G}$ as a graph signal $X^{(t)} \in \mathbb{R}^{N \times M}$ with $M$ being the number of features per node. We present the sequence of $T$ observations of graph signal as a three-dimensional tensor $\boldsymbol{X} \in \mathbb{R}^{T \times N \times M}$ s.t.

$$\boldsymbol{X}_t = [X^{(t)}, X^{(t+1)}, ..., X^{(t+T-1)}], \tag{1}$$

where $t$ is the initial time point of $T$ consecutive observations of graph signal. The problem of seizure detection is formulated as a binary classification task predicting the label $y \in \{0, 1\}$ for a corresponding tensor $\boldsymbol{X}_t$. The traffic forecasting problem is formulated as predicting the next tensor of $T$ consecutive observations of graph signal $\boldsymbol{X}_{t+T}$ from a given tensor $\boldsymbol{X}_t$.

## 3.2 SPATIAL AND TEMPORAL ENCODING

Let the input sequence in space and time be $x_n^t \in \mathbb{R}^M$, $n = 1, \ldots, N$, $t = 1, \ldots, T$, i.e., the $M$-dimensional feature vector for node $n$ of $\mathcal{G}$ at time step $t$. In Transformers terminology, we call $x_n^t$ the input token. We add spatial encoding to each token of the graph signal and we use the relative temporal encoding between two tokens at different time points $t, t'$. This allows the model to utilize both the position of the node in the graph and the time of the observation. The encodings are integrated into token $x_n^t$ after linear mapping as follows:

$$h_n^t = W_e x_n^t + z_n^{\text{spatial}}, \quad z_{tt'}^{\text{temporal}} = T_{tt'}(h_n^t, h_{n'}^{t'}) \tag{2}$$

where $h_n^t \in \mathbb{R}^P$ is the output feature after applying the spatial encoding and $W_e \in \mathbb{R}^{P \times M}$ is a linear mapping that transforms the input from $M$ features per node to $P$ features per node. The terms $z_n^{\text{spatial}}$ is the spatial encoding, which are applied to the input token $x_n^t$. $z_{tt'}^{\text{temporal}}$ is the relative temporal encoding which is applied relatively for two different observations of the graph at time point $t$ and $t'$.

**Temporal Encoding** To provide temporal encoding, we use relative temporal encoding as in (Wu et al., 2021). We encode the relative positions between input elements $h_n^t$ and $h_n^{t'}$ into trainable vectors $r_{tt'}^V, r_{t't}^Q, r_{tt'}^K \in \mathbb{R}^P$. $r_{tt'}^V, r_{t't}^Q, r_{tt'}^K$ are learnable vectors which are added to the attention matrix as relative temporal encoding and they are learned during training.

Importantly, the temporal embedding for all graph nodes at a given pair of time points $t, t'$ is uniform, as it is determined solely by the times of observation and not by the specific position of the node.

**Spatial Encoding** In order to capture the spatial information of node $n$ we define the spatial encoding as below:

$$z_n^{\text{spatial}} = U p_n. \tag{3}$$

Here, $p_n \in \mathbb{R}^2$ is the vector containing the positional information of node $n$ in the graph (comprising its $x$ and $y$ coordinates) and $U \in \mathbb{R}^{P \times 2}$ is the learnable weight matrix, $z_n^{\text{spatial}} \in \mathbb{R}^P$ is a vector of dimension $P$, which, along with temporal encoding, is added to the projected input. However, unlike the temporal encoding, the spatial encoding only carries information about the position of the node in the graph and does not depend on the time of the observation.

By combining the two encodings as shown in Equation (2), each token receives a unique spatial and relative temporal encoding that reflects both its spatial position in the graph and the time at which the signal was observed in relation to other time points. Ablation on the choice of spatial and temporal encodings, comparing them with other methods such as (Fuchs et al., 2020), are given in Appendix K. Our findings demonstrate that incorporating these spatial and temporal encodings significantly boosts performance, proving to be effective compared to scenarios where they are not used, as detailed in Appendix L. The motivation behind using $z_n^{\text{spatial}}$ is that in our settings the sensor locations are fixed, e.g., electrodes in EEG.

## 3.3 *NT* ATTENTION FORMULATION

After adding the encodings to each token we extract the key, query and value:

$$q_n^t = W_q h_n^t, \quad k_n^t = W_k h_n^t, \quad v_n^t = W_v h_n^t. \tag{4}$$

Here, $W_q, W_k, W_v \in \mathbb{R}^{P \times P}$ are weight matrices generating the query ($q_n^t$), key ($k_n^t$), and value ($v_n^t$) from $h_n^t$, where $P$ is the dimension of query, key, and values.

**Definition 3.1** (*NT* Attention). The attention score in *NT* Attention between the input tokens $x_n^t$ (the graph signal observation at node $n$ at time point $t$) and $x_{n'}^{t'}$ (the graph signal observation at node $n'$ at time point $t'$) is computed as

$$A_{n,n'}^{t,t'} = \text{softmax}\left(\frac{(q_n^t + r_{tt'}^Q)^\top (k_{n'}^{t'} + r_{tt'}^K)}{\sqrt{P}}\right). \tag{5}$$

From the equation above we compute the output $o_n^t \in \mathbb{R}^P$ of node $n$ at time $t$, based on its attention to all other tokens:

$$o_n^t = \sum_{n'=1}^{N} \sum_{t'=1}^{T} A_{n,n'}^{t,t'} (v_{n'}^{t'} + r_{tt'}^V). \tag{6}$$

Gathering all nodes and time steps, the output of the *NT*Attention module can therefore be represented as the tensor $\boldsymbol{O} \in \mathbb{R}^{T \times N \times P}$ with $\boldsymbol{O}_{tn} = o_n^t$. Unlike standard self-attention, our specific design allows all nodes of the graph at each time point to attend to each other (see Fig. 1). As commonly done in Transformers (Vaswani et al., 2017), we apply multiple heads of *NT*Attention by concatenating them.

### 3.4 GEOMETRY-AWARE ATTENTION MASKING (GMASK)

Masking the attention matrix can significantly enhance efficiency by skipping unnecessary computations and improving performance (Zaheer et al., 2020; Zhang et al., 2020). In this section, we introduce a geometry-aware masking approach, namely GMask, that aligns with the graph topology. We use the node positions in space characterized by $\mathcal{P}$ to create the following attention mask:

$$G_{ij} = \begin{cases} \exp\left(-\frac{\|p_i - p_j\|^2}{\sigma^2}\right) & \text{if } \|p_i - p_j\| \leq k, \\ 0 & \text{otherwise.} \end{cases}$$

Here, $\sigma$ is set to the standard deviation of the distances, and $k$ is the threshold for the Gaussian kernel (Shuman et al., 2013). We mask the attention by omitting the attention between two nodes $i$ and $j$ in all time points if $G_{ij} = 0$. This ensures that spatially distant nodes do not attend to each other, thereby improving the computational efficiency and accuracy of the model. Specifically, by masking the attention between nodes $i$ and $j$, $T^2$ entries in the attention matrix are set to zero because the model masks all the attention scores for all time points involving nodes $i$ and $j$. Therefore, let $N_0$ be the number of non-zero entries of $G$, GMask avoids $(N^2 - N_0)T^2$ floating point computations. We have also adapted dynamic GMask with correlation based edges (dynamic graph) in Appendix M. The structured sparsity induced by GMask aligns with the graph topology, focusing attention on relevant nodes and further enhancing optimization towards a solution that respects the underlying graph structure.

## 4 EMPIRICAL RESULTS

We evaluate *NT*Attention on two publicly available datasets for seizure detection and traffic forecasting. Capturing dependencies on both the spatial and temporal axes is crucial in these tasks, as detailed in Appendix H. Below we describe the results and data processing steps used for each task.

### 4.1 EEG-BASED SEIZURE DETECTION

**Dataset Preparation** We use the Temple University Hospital EEG Seizure Corpus (EEG) v.2.0.0 (Obeid & Picone, 2016; Shah et al., 2018), the largest publicly available EEG seizure database, which contains 5,545 EEG files for training, testing, and evaluation. These files are recorded using 19 EEG electrodes according to the standard 10-20 system (Jasper, 1958). Following previous studies (Ho & Armanfard, 2023; Tang et al., 2021; Asif et al., 2020), we segment the EEG signals into 1-second non-overlapping windows. For each window, we apply the Fourier transform and extract the log-amplitude of the frequency components, resulting in a graph signal $X^{(t)} \in \mathbb{R}^{N \times M}$, where $N = 19$ represents the EEG electrodes (nodes), and $M = 100$ represents the features per node. We then select $T$ consecutive observations to create an input EEG clip $\boldsymbol{X} \in \mathbb{R}^{T \times N \times M}$. Each clip is labeled $y = 1$ if it contains at least one seizure event and $y = 0$ if it does not. Dataset descriptions and additional preprocessing details are provided in Table 3 and Appendix A, respectively.

Table 5: **Seizure detection results**. Mean and standard deviations are from five random runs. Best mean results are highlighted in **bold**. All metrics are averaged using binary averaging.

| Clip Size | Model | AUROC | Weighted F1-Score | Sensitivity | Specificity |
|---|---|---|---|---|---|
| 12-s | Dense-CNN | $0.812_{\pm0.014}$ | $0.326_{\pm0.019}$ | $0.293_{\pm0.021}$ | $0.938_{\pm0.014}$ |
| | LSTM | $0.786_{\pm0.014}$ | $0.376_{\pm0.021}$ | $0.357_{\pm0.045}$ | $0.934_{\pm0.015}$ |
| | Transformer | $0.800_{\pm0.011}$ | $0.390_{\pm0.090}$ | $0.455_{\pm0.052}$ | $0.921_{\pm0.002}$ |
| | CNN-LSTM | $0.749_{\pm0.006}$ | $0.337_{\pm0.009}$ | $0.333_{\pm0.028}$ | $0.920_{\pm0.021}$ |
| | Corr-DCRNN | $0.812_{\pm0.012}$ | $0.392_{\pm0.027}$ | $0.373_{\pm0.035}$ | $0.935_{\pm0.012}$ |
| | Dist-DCRNN | $0.824_{\pm0.020}$ | $0.437_{\pm0.029}$ | $0.411_{\pm0.038}$ | $\mathbf{0.943_{\pm0.006}}$ |
| | REST | $0.834_{\pm0.012}$ | $0.437_{\pm0.22}$ | $0.391_{\pm0.04}$ | $0.912_{\pm0.08}$ |
| | *NT*Attention + GMask | $0.827_{\pm0.026}$ | $0.434_{\pm0.088}$ | $0.612_{\pm0.098}$ | $0.922_{\pm0.067}$ |
| | *NT*Attention | $\mathbf{0.842_{\pm0.021}}$ | $\mathbf{0.451_{\pm0.032}}$ | $\mathbf{0.638_{\pm0.081}}$ | $0.904_{\pm0.033}$ |
| 60-s | Dense-CNN | $0.796_{\pm0.014}$ | $0.404_{\pm0.022}$ | $0.451_{\pm0.134}$ | $0.869_{\pm0.071}$ |
| | LSTM | $0.715_{\pm0.016}$ | $0.365_{\pm0.009}$ | $0.463_{\pm0.060}$ | $0.814_{\pm0.053}$ |
| | Transformer | $0.781_{\pm0.100}$ | $0.372_{\pm0.092}$ | $0.442_{\pm0.001}$ | $0.878_{\pm0.055}$ |
| | CNN-LSTM | $0.682_{\pm0.003}$ | $0.330_{\pm0.016}$ | $0.363_{\pm0.044}$ | $0.857_{\pm0.023}$ |
| | Corr-DCRNN | $0.804_{\pm0.015}$ | $0.448_{\pm0.029}$ | $0.440_{\pm0.021}$ | $0.900_{\pm0.028}$ |
| | Dist-DCRNN | $0.793_{\pm0.022}$ | $0.341_{\pm0.170}$ | $0.326_{\pm0.183}$ | $0.932_{\pm0.058}$ |
| | REST | $0.782_{\pm0.123}$ | $0.441_{\pm0.072}$ | $0.388_{\pm0.01}$ | $0.924_{\pm0.702}$ |
| | *NT*Attention + GMask | $0.791_{\pm0.03}$ | $0.475_{\pm0.08}$ | $0.410_{\pm0.23}$ | $0.920_{\pm0.058}$ |
| | *NT*Attention | $\mathbf{0.810_{\pm0.021}}$ | $\mathbf{0.500_{\pm0.051}}$ | $\mathbf{0.489_{\pm0.102}}$ | $\mathbf{0.945_{\pm0.007}}$ |

**Time Window** Following (Saab et al., 2020; Tang et al., 2021), we evaluate the performance of our model and baselines for fast and slow seizure detection using $T = 12$ seconds and $T = 60$ seconds, respectively, as named in (Tang et al., 2021). Notably, time windows appear only in one EEG clip, and different clips do not share the same time window.

**Baselines** We use the following models as baselines for the seizure detection task: LSTM (Hochreiter & Schmidhuber, 1997), Dense-CNN (Saab et al., 2020), CNN-LSTM (Ahmedt-Aristizabal et al., 2020), two variations of DCRNN (Tang et al., 2021), REST (Afzal et al., 2024), and Transformer (Vaswani et al., 2017). Details of the baselines and their implementation are provided in Appendix C.

**Results** We evaluate the performance of *NT*Attention and various baselines using different metrics, including Area Under the Receiver Operating Characteristic Curve (AUROC), F1-Score, Sensitivity, and Specificity (Table 5). *NT*Attention demonstrates superior performance across all metrics, achieving particularly notable results in terms of Sensitivity. Specifically, *NT*Attention outperforms other benchmarks by around 5% points in F1-Score for slow detection and by 2% points in AUROC for fast seizure detection ($T = 12s$). Additionally, *NT*Attention is the only model to achieve a sensitivity above 50%, outperforming other benchmarks by about 20% points, a significant improvement for fast seizure detection. This high sensitivity is crucial for seizure detection tasks, as missing any seizure event can lead to potentially life-threatening situations for patients. Despite its high sensitivity, *NT*Attention also maintains competitive specificity, ensuring a balanced and effective detection performance. Furthermore, the vanilla transformer exhibits lower performance compared to the DCRNN baseline, suggesting its inability to effectively capture geometrical information. This highlights a key advantage of our model: the task-specific attention mechanism allows *NT*Attention to achieve high accuracy where the vanilla transformer falls short. Fig. 2 **b** visualizes the average attention scores over time for different EEG electrodes, applying various thresholds for GMask.

## 4.2 Traffic Forecasting

**Dataset Preparation** We used the METR-LA dataset (Jagadish et al., 2014), which contains traffic information collected from loop detectors on highways in Los Angeles County. This dataset provides a valuable resource for evaluating traffic forecasting models. Following previous studies (Li et al., 2018; Zheng et al., 2020; Shang et al., 2021), traffic speed data from 207 sensors was aggregated into 5-minute intervals and normalized using Z-score normalization. The data was split into 70% for training, 10% for evaluation, and 20% for testing (details provided in Table 4). Performance was measured across three forecasting horizons: 15 minutes (horizon 3), 30 minutes (horizon 6), and 1 hour (horizon 12) (Tang et al., 2021; Shang et al., 2021).

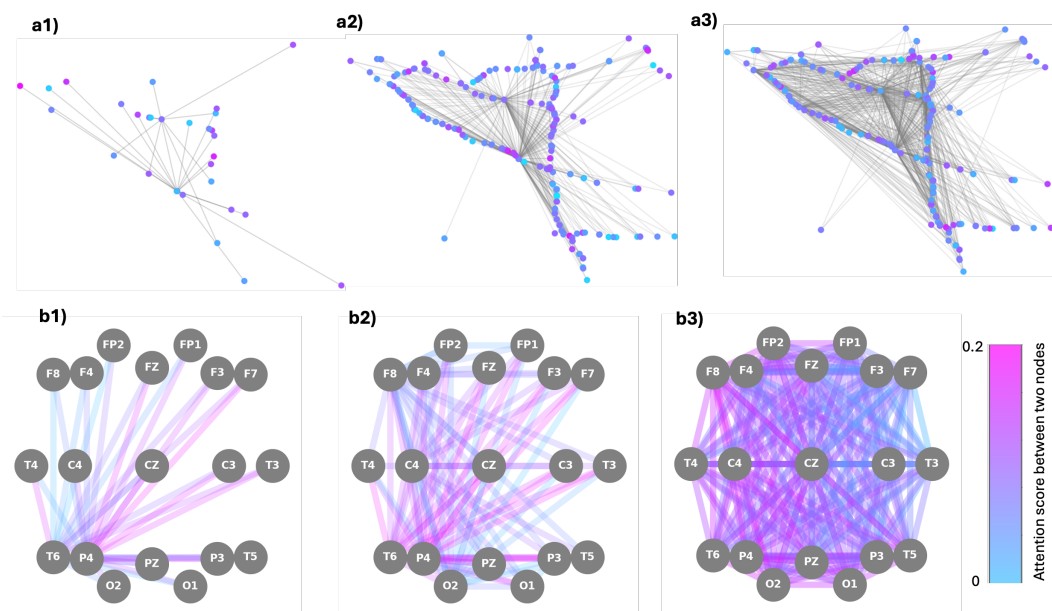

Figure 2: Visualization of attention scores between each pair of nodes, averaged over all time point pairs, for **a** traffic forecasting, and **b** seizure detection. For traffic forecasting, attention scores are shown with Gaussian kernel thresholds of **a1)** $k = 0.9$, **a2)** $k = 0.8$, and **a3)** $k = 0.5$. For seizure detection, **b1)** $k = 0.9$, **b2)** $k = 0.7$, and **b3)** $k = 0$ (no mask). The intensity of the attention scores is displayed within the range of 0 to 0.2 for seizure detection. For traffic forecasting, only attention scores higher than zero are visualized for better clarity given the large number of nodes.

**Baselines** We benchmarked *NT*Attention against several well-known traffic forecasting models, including HA (Historical Average), VAR (Hamilton, 2020), Support Vector Regression (SVR), Feed Forward Neural Network (FNN), LSTM (Hochreiter & Schmidhuber, 1997), DCRNN (Li et al., 2018), STGCN (Yu et al., 2018), GTS (Chen et al., 2021), ASTGN (Guo et al., 2019), PM-MemNet (Lee et al., 2021), STAEFormer (Liu et al., 2023), STDMAE (Gao et al., 2024) and GMAN (Zheng et al., 2020). Further details about baseline implementations are provided in the Appendix C.

**Results** As shown in Table 6, *NT*Attention achieves state-of-the-art performance with the lowest errors under both Root Mean Squared Error (RMSE) and Mean Absolute Error (MAE). Interestingly, unlike the seizure detection task, the accuracy of *NT*Attention increases with GMask, suggesting that in a larger network it is beneficial to exactly nullify the attention scores of spatially distant nodes.

Fig. 2 **a** visualizes the attention between different nodes for traffic forecasting task, highlighting how attention varies with and without masking. Notably, for longer forecasting horizons, such as the 12-step horizon, our model achieves a significant improvement with an MAE of 2.93 and an RMSE of 5.82, which is considerably lower than all other baselines for traffic forecasting. This demonstrates the effectiveness of *NT*Attention in capturing long-range space and time dependencies and improving predictive accuracy over extended periods.

### 4.3 COMPARISON BETWEEN DIFFERENT MASKING STRATEGIES

We also compare Geometry Aware Masking (GMask) with other well-known masking strategies, including Random Masking (Peng et al., 2021), Window Masking (Beltagy et al., 2020), and BGBIRD (Zaheer et al., 2020), on the traffic forecasting task for Horizon 3 as shown in Table 7. Figure 3 visualizes the different masking strategies. Our observations indicate that GMask achieves the highest accuracy among all other types of masking. This superiority is attributed to GMask's design, which aligns with the graph geometry, ensuring that node connections are respected. In contrast, other types of masking either neglect node connections by masking randomly or focus too closely on the diagonal, resulting in lower accuracy. Random Masking, for instance, can lead to spatially distant

Table 6: **Trafic forecasting results**. Lowest MAE and RMSE errors are highlighted in **bold**.

| Model | Horizon 3 | | Horizon 6 | | Horizon 12 | |
|---|---|---|---|---|---|---|
| | MAE | RMSE | MAE | RMSE | MAE | RMSE |
| HA | 4.79 | 10.00 | 5.47 | 11.45 | 6.99 | 13.89 |
| VAR | 4.42 | 7.8 | 5.41 | 9.13 | 6.52 | 10.11 |
| SVR | 3.39 | 8.45 | 5.05 | 10.87 | 6.72 | 13.76 |
| FNN | 3.99 | 8.45 | 4.23 | 8.17 | 4.49 | 8.69 |
| LSTM | 3.44 | 6.3 | 3.77 | 7.23 | 4.37 | 8.69 |
| DCRNN | 2.77 | 5.38 | 3.47 | 7.24 | 4.59 | 9.4 |
| GTS | 2.67 | 5.27 | 3.04 | 6.25 | 3.46 | 7.31 |
| ASTGN | 4.86 | 9.27 | 5.43 | 10.61 | 6.51 | 12.52 |
| GMAN | 2.8 | 5.55 | 3.12 | 6.49 | 3.44 | 7.35 |
| STAEFormer | 2.65 | 5.11 | 2.97 | 6.00 | 3.34 | 7.62 |
| STDMAE | **2.62** | 5.62 | 2.99 | 7.47 | 3.4 | 7.07 |
| PM-MemNet | 2.66 | 5.28 | 3.02 | 6.28 | 3.4 | 7.24 |
| STGCN | 2.88 | 5.47 | 3.07 | 6.22 | 3.53 | 7.37 |
| *NT*Attention | 2.92 | 5.63 | 2.68 | 6.04 | 3.21 | 6.44 |
| *NT*Attention + GMask | 2.64 | **4.34** | **2.50** | **4.37** | **2.93** | **5.82** |

nodes attending to one another, while GMask effectively captures the graph structure, resulting in the lowest MAE and RMSE among all strategies. In Appendix J, we theoretically examine various masking strategies and models used for the theoretical complexity of traffic forecasting. Additionally, details about the masking implementation are provided in Appendix E.

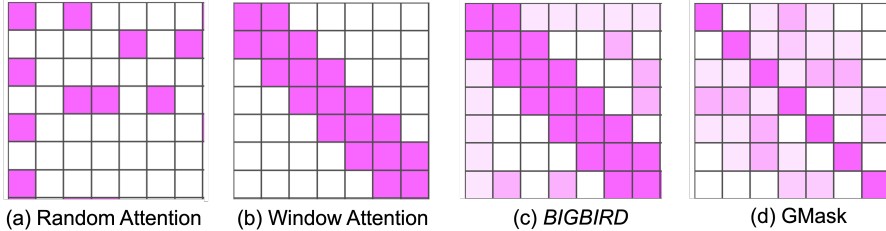

(a) Random Attention    (b) Window Attention    (c) *BIGBIRD*    (d) GMask

Figure 3: Illustration of different masking strategies **a)** Random with $r = 2$ **b)** Window attention with $w = 3$ **c)** *BIGBIRD* **d)** GMask (ours). White color indicate absence of attention.

Table 7: Comparison of our proposed GMask to other masking strategies for *NT*Attention on traffic forecasting for Horizon 3. Lowest MAE and RMSE errors are highlighted in **bold**.

| Model | MAE | RMSE |
|---|---|---|
| No Mask | 2.9 | 5.6 |
| RandomMask | 7.4 | 14.5 |
| WindowMask | 5.3 | 9.67 |
| *BIGBIRD* | 4.32 | 8.02 |
| GMask | **2.64** | **4.34** |

Table 8: Training time for one epoch of *NT*Attention with and without GMask on the TUSZ and METR-LA datasets. Lowest training time are highlighted in **bold**.

| Model | Dataset | Training time |
|---|---|---|
| W/O Mask | METR-LA | 10 min - 30 sec |
| W/O Mask | TUSZ | 1min -5 sec |
| W/ + GMask | METR-LA | **7 min - 20sec** |
| W/ + GMask | TUSZ | **20 sec** |

### 4.4 EFFICIENCY ANALYSIS

We compare the computational efficiency of various methods for seizure detection and traffic forecasting tasks. *NT*Attention has complexity of $\mathcal{O}(N^2T^2)$, where after applying GMask, scalability with respect to the number of nodes becomes linear, with $\mathcal{O}(\alpha(N)NT^2)$, where $\alpha(N)$ is the number of nonzero neighbors for each node. Details of training time for *NT*Attention with and without masking are provided in Appendix D, where we show that GMask dramatically improves efficiency while maintaining a similar level of performance. In Figure 4, we compare number of parameters, FLOPS, and model size. *NT*Attention + GMask achieves competitive number of FLOPs maintaining

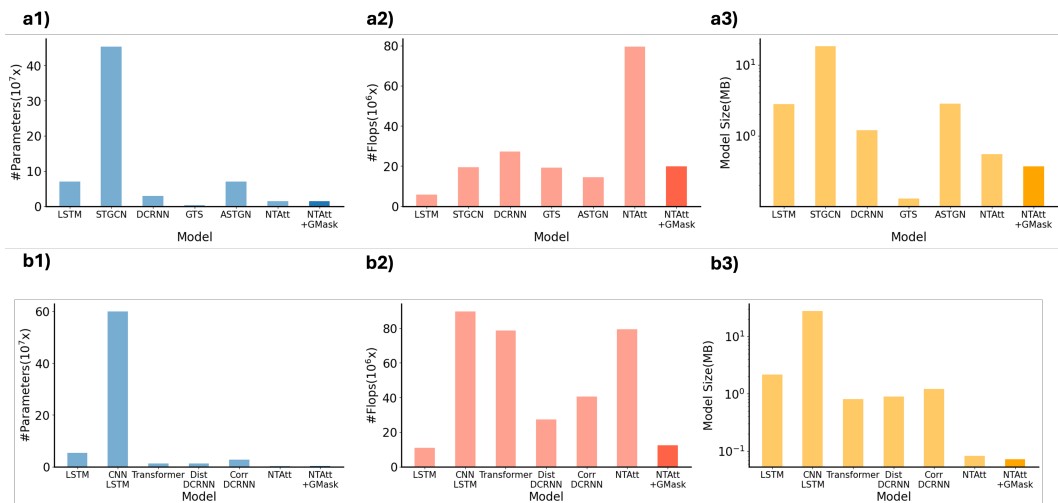

Figure 4: Comparison of Model Efficiency Across Tasks: (**1**) Number of Parameters, (**2**) FLOPs, and (**3**) Model Size for (**a**) Traffic Forecasting and (**b**) Seizure Detection Tasks.

low number of parameters and small model size, at the same time achieving SOTA accuracy. This advantage arises from the specific attention mechanism used in *NT*Attention, which enables rich information to flow through all time steps and nodes, allowing for effective decoding of graph signals with considerably fewer parameters than other benchmarks, particularly RNN-based models. The application of GMask effectively sparsifies the attention matrix and significantly reduces the number of FLOPs, as demonstrated in Figure 4, and achieving notably higher accuracy. Numerical comparisons are reported in Appendix F. More details on the efficiency improvements through the sparsification via GMask in Appendix G.

**Effect of GMask on Seizure Detection vs. Traffic Forecasting:** We observed that GMask improves both efficiency and performance in traffic forecasting, while in seizure detection, it enhances efficiency but may slightly reduce performance. This difference arises because the EEG graph in seizure detection consists of only 19 nodes, as shown in Fig. 2 **b**. Sparsifying with GMask can lead to imbalanced masking (**b2, b1**), and with so few nodes, the attention mechanism can effectively capture dynamics without additional masking. Additionally, as shown in Fig. 4, the efficiency benefits of GMask are less significant in this case. In contrast, for larger graphs like those in traffic forecasting, GMask reduces computational complexity and filters out unwanted attention between distant nodes, leading to both improved accuracy and efficiency. This highlights the effectiveness of GMask in scenarios with larger graphs.

## 5 CONCLUSION

In this study, we proposed *NT*Attention for graph signal processing tasks, specifically targeting seizure detection and traffic forecasting. By incorporating spatial encoding into each node's features and relative temporal encoding into the attention matrix, we effectively utilized the positional and temporal information inherent in the data. Our space-time attention mechanism, enhanced with geometry-aware masking based on graph topology, further improved model performance by focusing attention on relevant nodes. We evaluated *NT*Attention on the TUH EEG seizure dataset and the METR-LA traffic dataset, benchmarking it against several well-known models. For seizure detection, *NT*Attention demonstrated superior performance, achieving significantly higher F1-scores and sensitivity than other baselines, emphasizing its ability to reliably detect seizure events. In traffic forecasting, *NT*Attention achieved the lowest RMSE and MAE, particularly excelling in long-term forecasting horizons. *NT*Attention requires similar memory and computations as the baselines, where efficiency can be further boosted with GMask while maintaining a similar level of performance.

## 6 ETHICAL STATEMENT FOR TUSZ DATASET

The EEG Seizure Corpus from Temple University Hospital, utilized in our research, is anonymized and publicly accessible with IRB approval Obeid & Picone (2016); Shah et al. (2018). The authors declare no conflicts of interest, and the seizure detection models presented in this study do not provide any harmful insights. Also, dataset is publicly available anonymously for all patients.

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

## A    DETAILS OF DATA PROCESSING

### A.1    EEG BASED SEIZURE DETECTION

Due to varying sampling frequencies in the Temple University EEG Seizure Corpus (TUSZ), we standardize the signals to a uniform frequency of 200 Hz. We then preprocess the data to generate EEG clips in the frequency domain along with their corresponding labels. For seizure detection, we utilize both seizure and non-seizure EEGs. EEG clips are created by sliding a 12-second (or 60-second) window over the signals with no overlap, discarding the last window if it is shorter than the clip length. Each clip is labeled as $y = 1$ if it contains at least one seizure event, and $y = 0$ if no seizure event is present. For each 1-second window, we applied FFT to extract frequency domain features and selected the log-amplitude of non-negative frequency samples, resulting in $X^{(t)} \in \mathbb{R}^{N \times M}$ matrices for each 1-second window, with $N = 19$ EEG channels and $M = 100$ frequency features (Tang et al., 2021; Ho & Armanfard, 2023). We then applied Z-score normalization to each window. For the seizure detection task, we used $T = 12$-second (or 60-second) consecutive windows, resulting in an EEG clip tensor $\boldsymbol{X} \in \mathbb{R}^{T \times N \times M}$ for each input, with a corresponding label $y$. Due to the imbalance between the number of seizure and normal samples, we down sampled the normal samples during training to ensure an equal number of seizure and non-seizure samples. However, all samples were used for testing. Additionally, following the methodology in (Tang et al., 2021), we divided the TUSZ dataset's training set into a 90-10 ratio for training and validation.

### A.2    TRAFFIC FORECASTING

Following previous studies (Li et al., 2018; Zheng et al., 2020; Shang et al., 2021; Guo et al., 2019), we processed the METR-LA (Tang et al., 2021) dataset by selecting 207 traffic sensors and aggregating the traffic sensor readings into 5-minute windows, resulting in $X^{(t)} \in \mathbb{R}^{N \times M}$ tensors with $N = 207$ nodes and $M = 2$ features per node for each input sample. We then selected $T = 3$ consecutive windows for 15-minute forecasting (Horizon 3), $T = 6$ for 30-minute forecasting (Horizon 6), and $T = 12$ for one-hour forecasting (Horizon 12), resulting in an input tensor $\boldsymbol{X} \in \mathbb{R}^{T \times N \times M}$.

We split the data into 70% for training, 20% for testing, and 10% for evaluation. Z-score normalization was applied to each 5-minute window of data. Additionally, each input tensor $\boldsymbol{X}$ overlaps with the previous input tensor for $T - 1$ time windows, with only one new window differing between consecutive input tensors. During training, teacher-forcing was applied for all models, while during testing, the models were forced to predict based on their previous predictions.

## B    *NT*ATTENTION CONFIGURATION DETAILS

As depicted in Fig. 1, our model employs multi-head attention and a fully connected network, similar to the vanilla Transformer architecture (Vaswani et al., 2017). The fully connected network is defined as follows:

$$\text{FFN}(x) = \max(0, xW_1 + b_1)W_2 + b_2 \tag{7}$$

Here, $x$ is the input, $W_1$ and $W_2$ are weight matrices, $b_1$ and $b_2$ are bias vectors, and $\max(0, \cdot)$ denotes the ReLU activation function. The full configuration of *NT*Attention model for both tasks is provided in Tables 9.

## C    BASELINES

### C.1    SEIZURE DETECTION

**DCRNN** We adhered to the hyperparameter tuning strategy from the original paper (Tang et al., 2021) for both standard DCRNN and the self-supervised variant. The hyperparameter search on the validation set included: a) Initial learning rate in the range [5e-5, 1e-3]; b) Number of Diffusion Convolutional Gated Recurrent Units (DCGRU) layers in the range {2, 3, 4, 5} and hidden units in the range {32, 64, 128}; c) Maximum diffusion step $K$ in {2, 3, 4}; d) Dropout probability in the

Table 9: Model Configurations and Hyper-parameters of *NT*Attention.

|  | Seizure detection | Traffic Forecasting |
| --- | --- | --- |
| **# Total param** | 21.3K | 275K |
| **#Layers** | 2 | 2 |
| **Hidden Dimension $P$** | 32 | 32 |
| **FFN Inner-layer Dimension** | 32 | 32 |
| **Input projection dimension** | 32 | 32 |
| **#Attention Heads** | 16 | 16 |
| **Hidden Dimension of Each Head** | 32 | 32 |
| **FFN Dropout** | 0.1 | 0.1 |
| **Attention Dropout** | 0.1 | 0.1 |
| **Max Epochs** | 30 | 20 |
| **Geometry-aware mask threshold** | 0.0 (best result with no mask) | 0.5 |
| **Peak Learning Rate** | 1e-3 | 1e-3 |
| **Batch Size** | 128 | 64 |
| **Learning Rate Decay** | Cosine (Loshchilov & Hutter, 2016) | Cosine (Loshchilov & Hutter, 2016) |
| **Adam $\epsilon$** | 1e-8 | 1e-8 |
| **Adam ($\beta_1, \beta_2$)** | (0.9, 0.999) | (0.9, 0.999) |
| **Weight Decay** | 0.0 | 0.0 |
| **Last layer dimension** | 1 | 207 |

final fully connected layer. e) Two variations of the model, one utilizing a correlation-based graph and the other a distance-based graph, were implemented as described in (Tang et al., 2021). Models were trained for 50 epochs with an initial learning rate of 5e-4, using a maximum diffusion step of 1 and 64 hidden units in both the encoder and decoder. Additionally, we employed a cosine annealing learning rate scheduler (Loshchilov & Hutter, 2016).

**CNN-LSTM**: For the CNN-LSTM baseline, we used the model architecture specified in (Ahmedt-Aristizabal et al., 2020). This configuration includes two stacked convolutional layers with 32 kernels of size 3×3, one max-pooling layer of size 2×2, one fully connected layer with 512 output neurons, two stacked LSTM layers with a hidden size of 128, and an additional fully connected layer.

**Dense-CNN**: Dense-CNN, we employ the same model architecture as that described in (Saab et al., 2020).

**LSTM**: We used two stacked RNN layers, each with 64 hidden units, followed by a fully connected layer for the final prediction.

**Transformer**: We used two layer transformer with original positional encoding for different time points.

**REST**: A graph-based RNN using residual update for updating its state designed for seizure detection and classification task. Implementation are followed by Afzal et al. (2024) which includes 2 layers of update cell with 32 neourons for projection.

## C.2 TRAFFIC FORECASTING

**HA**: The Historical Average (HA) model, as described in (Tang et al., 2021), predicts traffic flow by averaging historical data over a one-week period, providing stable performance regardless of short-term changes in the forecasting horizon.

**VAR:** Implemented using the statsmodel python package, the Vector Auto-regressive model (Hamilton, 2020) sets the number of lags to 3.

**SVR:** Utilizing Linear Support Vector Regression with a penalty term C of 0.1, this model considers the 5 most recent historical observations.

**FNN:** A Feed forward neural network with two hidden layers, each containing 256 units. It employs an initial learning rate of 1e-3, reducing to 1/10 every 20 epochs after the 50th epoch. Dropout with a ratio of 0.5 and L2 weight decay of 1e-2 are applied to all hidden layers, with training using batch size 64 and MAE as the loss function. Early stopping is triggered by monitoring the validation error (Tang et al., 2021; Shang et al., 2021).

**FC-LSTM:** This Encoder-decoder framework utilizes LSTM with peephole (Sutskever et al., 2014), incorporating two recurrent layers in both the encoder and decoder. Each layer consists of 256 LSTM units, with L1 weight decay set to 2e-5 and L2 weight decay to 5e-4. Training involves batch size 64 and MAE as the loss function, with an initial learning rate of 1e-4, reducing to 1/10 every 10 epochs after the 20th epoch. Early stopping is implemented based on the validation error (Tang et al., 2021; Zheng et al., 2020).

**DCRNN:** The Diffusion Convolutional Recurrent Neural Network comprises two recurrent layers in both the encoder and decoder, each with 64 units. Model description and details of implementation is followed by original paper (Tang et al., 2021).

**ASTGCN**: ASTGCN integrates the spatial-temporal attention mechanism to capture dynamic spatial-temporal characteristics simultaneously (Guo et al., 2019).

**GMAN**: GMAN is an attention-based model that employs spatial, temporal, and transform attentions in stacked layers (Zheng et al., 2020).

**STGCN**: STGCN is a type of GNN leveraging graph-based convolution structures to capture comprehensive spatio-temporal correlations in traffic flow data (Yu et al., 2018).

**GTS**: GTS learns a graph structure among multiple time series and simultaneously forecasts them using DCRNN (Shang et al., 2021).

**PM-MemNet**: Pattern-Matching Memory Networks (PM-MemNet) learn to match input data to representative patterns using a key-value memory structure (Lee et al., 2021).

**STAForemr**: A spatial and temporal attention mechanism that provides separate attention for each domain, implemented as described in Liu et al. (2023).

**STDMAE**: Self-supervised pre-training framework STD-MAE uses two decoupled masked autoencoders to reconstruct spatiotemporal series along spatial and temporal dimensions. the implementation followed by Gao et al. (2024).

All models for both tasks were trained on single NVIDIA A100 GPU.

## D    TRAINING TIME

Table 10: Training time for one epoch of *NT*Attention with and without GMask on the TUSZ and METR-LA datasets. Lowest training time are highlighted in **bold**.

| Model | Dataset | Batch | Widow size - Clip length | # Nodes | Training time |
|---|---|---|---|---|---|
| *NT*Attention | METR-LA | 64 | $T = 3$ (Horizon 3) | 207 | 10 min - 30 sec |
| *NT*Attention | TUSZ | 128 | $T = 12$ (Fast detection) | 19 | 1min -5 sec |
| *NT*Attention + GMask | METR-LA | 64 | $T = 3$ (Horizon 3) | 207 | **7 min - 20sec** |
| *NT*Attention + GMask | TUSZ | 128 | $T = 12$ (Fast detection) | 19 | **20 sec** |

## E    MASKING DETAILS

All the hyperparameters for different masking strategies used in seizure detection and traffic forecasting tasks were tuned on the validation set. The specific hyperparameters for each masking strategy are as follows:

**Random Mask**: The parameter $r$ was tuned within the interval $\{5, 10, 15\}$ for seizure detection and $\{10, 30, 50, 100, 150\}$ for traffic forecasting.

**Window Attention**: The parameter $w$ was tuned within the interval $\{1, 3, 5\}$ for seizure detection and $\{5, 10, 20\}$ for traffic forecasting.

*BIGBIRD*: Parameters $w$ and $r$ were tuned similarly, with $g$ tokens attending all parts of the sequence lying in $\{3, 5, 10\}$ for both tasks.

**GMask**: The threshold parameter $k$ was tuned within $\{0.5, 0.7, 0.9\}$.

Additionally, all the masks were applied to the nodes in the same manner as described in Section 3.4 for GMask, and they were not used for masking temporal information.

## F  DETAILED EFFICIENCY AND ACCURACY COMPARISON

Table 11: Comparison of Efficiency and Training Time for Models in the Traffic Forecasting Task (First and Second best are bold)

| Model | #Parameters | #FLOPs | Model Size (MB) | Train-Time/Epoch (s) | Average MAE |
|---|---|---|---|---|---|
| FNN | $2.8 \times 10^9$ | $5.6 \times 10^9$ | 30.1 | 180.43 | 5.25 |
| LSTM | 695,808 | 5,700,608 | 2.80 | 31.3 | 3.86 |
| STGCN | 454,000 | 19,347,840 | 18.2 | 284.5 | 3.16 |
| DCRNN | 297,339 | 27,278,592 | 1.20 | 691.32 | 3.61 |
| GTS | 32,291 | 19,289,088 | 0.13 | 105 | 5.14 |
| ASTGN | 705,315 | 14,386,176 | 2.83 | 303.2 | 5.6 |
| NTAtt | 141,800 | 79,541,504 | 0.584 | 633.5 | 3.13 |
| NTAtt+GMask | 141,800 | 19,951,488 | 0.368 | 442.3 | 2.73 |

Table 12: Comparison of Efficiency and Training Time for Models in the Seizure Detection Task (First and Second best are bold)

| Model | #Parameters | #FLOPs | Model Size (MB) | Train-Time/Epoch (s) | Average AUROC |
|---|---|---|---|---|---|
| LSTM | 536,000 | 10,976,522 | 2.147 | 4.2 | 75.1 |
| CNNLSTM | 6,000,000 | 89,762,122 | 27.6 | 6 | 71.5 |
| Transformer | 123,000 | 78,654,123 | 0.8 | 12 | 79.05 |
| DistDCRNN | 126,000 | 27,278,592 | 0.884 | 30 | 80.8 |
| CorrDCRNN | 264,000 | 40,557,184 | 1.2 | 35 | 80.2 |
| NTAtt | 21,300 | 79,541,504 | 0.083 | 45 | 80.2 |
| NTAtt+GMask | 21,300 | 12,366,021 | 0.273 | 20 | 82.4 |

## G  SCALABILITY AND SPARSITY OF GMASK

We conducted an ablation study to examine the effect of the threshold parameter $k$ on the number of FLOPs for *NT* Attention + GMask. Our findings indicate that, in graphs with a larger number of nodes, such as traffic sensors, the threshold parameter exponentially decreases the number of connections, resulting in a sparser attention matrix and reduced computational requirements for the model (Figure 5 **b**). Additionally, we observed a significant reduction in the number of FLOPs for EEG signals, although this reduction is more linear compared to larger graphs. This demonstrates how *NT* Attention + GMask achieves comparable model size and efficiency to other benchmarks, as the attention matrix becomes increasingly sparse.

## H  MOTIVATION FOR LONG-RANGE INTERACTIONS

We provide detailed examples from two key tasks:

**Seizure Detection:**

- **Long-range spatial dependency:** Seizures exhibit significant variability in their characteristics. For example, focal seizures may manifest at a single electrode while other electrodes show normal rhythms. For effective detection, a model must enable message passing across distant nodes. Without this capability, such seizures might be missed if they occur in isolation or detected too late, as traditional graph neural networks often require multiple time windows to propagate messages across the network.
- **Long-range temporal dependency:** In 60-second, 250Hz windows, if a seizure occurs at the beginning with no subsequent activity, the model must utilize long-range temporal reasoning to accurately classify the window based on this brief episode.

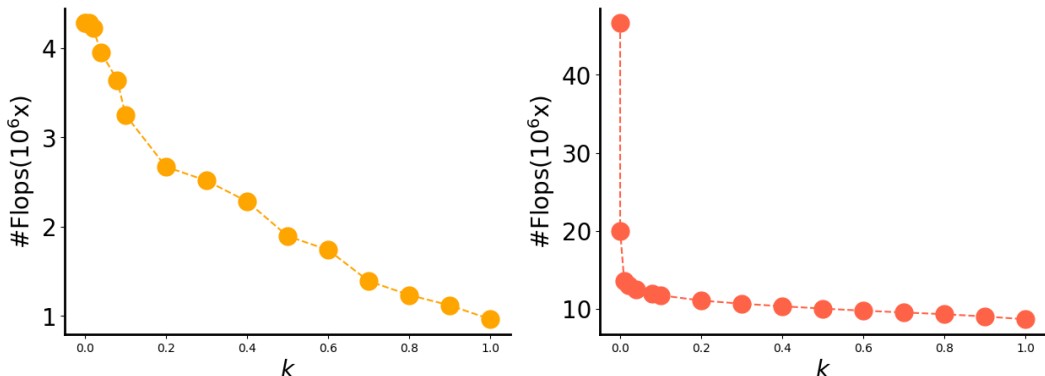

Figure 5: Number of FLOPs of *NT*Attention +GMask for **(Left)** Seizure detection task with 19 nodes and **(Right)** Traffic forecasting task with 207 nodes, based on the Gaussian threshold chosen for GMask (representing the sparsity of the mask).

**Traffic Forecasting:**

- **Long-range temporal dependency:** Traffic forecasting typically operates with a 5-minute window resolution. Forecasting 1 or 2 hours ahead involves processing 12 or 24 data windows, which is considered long-range for this task. Even forecasting for 1 hour with short windows necessitates managing long sequences of data.

- **Long-range spatial dependency:** Congestion at a major node can create ripple effects, impacting distant secondary nodes due to rerouted vehicles, changes in traffic signals, and altered driver behavior. Understanding these long-range spatial dynamics can significantly enhance forecasting performance.

## I  EXTENSION OF *NT*ATTENTION TO GNNS

The spatial-temporal nature of the data we are using requires both spatial and temporal dependencies, which cannot be handled by traditional GNNs alone. This is why the community is adapting methods like GMAN (Zheng et al., 2020), and GTS (Shang et al., 2021) for such tasks.

We would like to emphasize that *NT*Attention can be viewed as an extension of current GNNs, particularly graph transformers like Graphormer (Ying et al., 2021), to temporal graphs. Traditional GNNs lack the capability to handle time-series based features, making them unsuitable for the type of temporal data and tasks we address.

In contrast, models such as DCRNN Tang et al. (2021); Li et al. (2018) have been adapted to this setting, utilizing a combination of GConv (Morris et al., 2019) and GRU to capture spatio-temporal dependencies. *NT*Attention extends these ideas by integrating temporal features directly into the graph-based model, enhancing its ability to process and analyze temporal graph data effectively.

## J  THEORETICAL COMPLEXITY OF MODELS

We compare the theoretical complexities of different methods and their capabilities w.r.t. space and time learning in Table 13. Naive *NT*Attention scales quadratically with the addition of new nodes and time steps, having a complexity of $O(N^2 T^2)$. However, after applying GMask, scalability with respect to the number of nodes becomes linear, with a complexity of $O(\alpha(N)NT^2)$. Here, $\alpha(N)$ is the number of non-zerod neighbour nodes and depends graph topology. Figure 5 shows that applying GMask, even with small $k$ values, exponentially reduces computational complexity, especially for large graphs like traffic data.

Table 13: Comparison of theoretical complexity of models

| Model | Complexity | Spatial | Temporal |
|-------|-----------|---------|----------|
| Naive Transformer | $O(T^2)$ | ✗ | ✔ |
| Graph Transformer, GNN | $O(N^2)$ | ✔ | ✗ |
| STTN/ATGCN | $O(N^2 + T^2)$ | ✔ | ✔ |
| BigBird | $O(r \cdot T)$ | ✗ | ✔ |
| Window Attention | $O(w \cdot T)$ | ✗ | ✔ |
| *NT*Attention | $O(N^2 T^2)$ | ✔ | ✔ |
| *NT*Attention +GMask | $O(\alpha(N) N T^2)$ | ✔ | ✔ |

## K ABLATION ON CHOICE OF SPATIAL AND TEMPORAL ENCODING

Tables 14 and 15 present a comparison of various spatial and temporal encoding methods for seizure detection and traffic forecasting, examining their impact on performance metrics. The methods evaluated include Fixed Temporal Encoding (Fixed-TE), Rotational Spatial Encoding (Rot-SE), Relative Temporal Encoding (Rel-TE), and Fixed Spatial Encoding (Fixed-SE). These results highlight the contribution of the components in *NT*Attention.

Table 14: Evaluation of various encodings for seizure detection, including Fixed Temporal Encoding (Fixed-TE), Rotational Spatial Encoding (Rot-SE), Relative Temporal Encoding (Rel-TE), and Fixed Spatial Encoding (Fixed-SE).

| Clip Size | Model | AUROC | F1-Score | Sensitivity | Specificity |
|-----------|-------|-------|----------|-------------|-------------|
| 12-s | Fixed-TE + Fixed-SE | 0.84 | 0.450 | 0.633 | 0.902 |
|  | Rot-SE + Rel-TE | 0.83 | 0.461 | 0.667 | 0.812 |
|  | Fixed-TE+Rot-SE | 0.84 | 0.444 | 0.630 | 0.870 |
|  | Fixed-TE+Rot-SE+GMask | 0.84 | 0.444 | 0.630 | 0.870 |
|  | *NT*Attention + GMask | 0.827 | 0.434 | 0.612 | **0.922** |
|  | *NT*Attention | **0.842** | **0.451** | **0.638** | 0.904 |
| 60-s | Fixed-TE + Fixed-SE | 0.782 | 0.471 | 0.421 | 0.921 |
|  | Rot-SE + Rel-TE | 0.771 | 0.500 | 0.410 | 0.782 |
|  | Fixed-TE+Rot-SE | 0.784 | 0.541 | 0.400 | 0.927 |
|  | Fixed-TE+Rot-SE+GMask | 0.782 | 0.613 | 0.511 | 0.843 |
|  | *NT*Attention + GMask | 0.791 | 0.475 | 0.410 | 0.920 |
|  | *NT*Attention | **0.810** | **0.671** | **0.489** | **0.945** |

Table 15: Evaluation of various encodings for traffic forecasting, including Fixed Temporal Encoding (Fixed-TE), Rotational Spatial Encoding (Rot-SE), Relative Temporal Encoding (Rel-TE), and Fixed Spatial Encoding (Fixed-SE).

| Model | H3 MAE | H3 RMSE | H6 MAE | H6 RMSE | H12 MAE | H12 RMSE |
|-------|--------|---------|--------|---------|---------|----------|
| Fixed-TE + Fixed-SE | 2.90 | 5.60 | 2.70 | 6.00 | 3.80 | 7.21 |
| Rot-SE + Rel-TE | 2.8 | 5.3 | 2.78 | 4.84 | 3.1 | 6.32 |
| Fixed-TE+Rot-SE | 3.00 | 5.76 | 3.22 | 6.00 | 3.67 | 7.78 |
| Fixed-TE+Rot-SE+GMask | 2.63 | 4.35 | 2.6 | 4.52 | 2.98 | 5.80 |
| *NT*Attention | 2.92 | 5.63 | 2.68 | 6.04 | 3.21 | 6.44 |
| *NT*Attention + GMask | **2.64** | **4.34** | **2.50** | **4.37** | **2.93** | **5.82** |

## L ABLATION REMOVING THE SPATIO-TEMPORAL ENCODINGS

Table 16 presents the traffic forecasting performance when spatial and temporal encodings are removed from the model in various configurations. The impact of *NT*Attention and its enhanced version with GMask is also highlighted, showing the best performance across all metrics.

Table 16: Traffic forecasting results without spatial and temporal encodings.

| Model | H3 MAE | H3 RMSE | H6 MAE | H6 RMSE | H12 MAE | H12 RMSE |
|---|---|---|---|---|---|---|
| W/O TE | 3.56 | 7.54 | 4.21 | 9.12 | 6.80 | 12.70 |
| W/O SE | 3.78 | 7.66 | 4.50 | 9.50 | 6.52 | 13.02 |
| W/O SE+TE | 3.89 | 7.78 | 4.56 | 9.89 | 6.99 | 13.89 |
| *NT*Attention | 2.92 | 5.63 | 2.68 | 6.04 | 3.21 | 6.44 |
| *NT*Attention + GMask | **2.64** | **4.34** | **2.50** | **4.37** | **2.93** | **5.82** |

## M   *NT*ATTENTION WITH DYNAMIC GRAPH

Extending *NT*Attention to handle dynamic edges involves adapting the GMask based on temporal correlations between nodes, which aligns better with the data and is less computationally intensive as node positions remain unchanged. Instead of spatial distances, GMask can be generated using node feature correlations over time, as explored in studies like Tang et al. (2021). This approach is formulated as:

$$G_{ij} = \text{corr}(x_i^t, x_j^t)$$

where $x_i^t$ and $x_j^t$ are the feature vectors of nodes $i$ and $j$ at time $t$, and $\text{corr}(x_i^t, x_j^t)$ is the correlation coefficient between them.

We have conducted an ablation study to explore how dynamic GMask impacts traffic forecasting results as shown below:

Table 17: Impact of Dynamic GMask on Traffic Forecasting Results

| Model | H3 MAE | H3 RMSE | H6 MAE | H6 RMSE | H12 MAE | H12 RMSE |
|---|---|---|---|---|---|---|
| *NT*Attention | 2.92 | 5.63 | 2.68 | 6.04 | 3.21 | 6.44 |
| *NT*Attention +Dynamic GMask | **2.64** | 5.53 | 2.67 | 5.87 | 3.40 | 6.50 |
| *NT*Attention + GMask | **2.64** | **4.34** | **2.50** | **4.37** | **2.93** | **5.82** |

## N   RATIO OF GMASK VS RANDOM MASK

We have analyses the performance of GMask vs the random Mask strategy on different $k$ thresholds which shows that GMask is superior and more aligned with the graph nature of data compared to Random Mask in all different thresholds.

Table 18: GMask vs Random Performance

| Mask | $k = 0.1$ | $k = 0.5$ | $k = 0.9$ |
|---|---|---|---|
| GMask | **5.45** | **4.34** | **5.21** |
| Random | 17.2 | 14.5 | 12.22 |

