# OpenReview forum: "Space-time self-attention for graph signal processing"
_ICLR.cc/2025/Conference — Submitted to ICLR 2025_

### Official Review · Reviewer_1iQN · 2024-11-02

**Soundness:** 2
**Presentation:** 3
**Contribution:** 2
**Rating:** 5
**Confidence:** 3

**Summary:**

The paper proposed NTAttention for graph signal processing tasks, specifically targeting
seizure detection and traffic forecasting. By incorporating spatial encoding into each node’s features
and relative temporal encoding into the attention matrix, NTAttention provides a new way to learning with spatial-temporal data.

**Strengths:**

1. Developing a representation learning framework that could integrate both spatial and temporal information from the data is an important and meaningful learning problem.

2. The proposed method achieves good performance on the seizure detection and traffic forecasting tasks.

**Weaknesses:**

Overall, I think the empirical evaluation should be more solid, to provide more understanding of the problem and the proposed method:

1. Lack of intuition behind the method: it is quite unclear to me why the proposed way to encode spatial information in the node embedding, then add in the temporal information in the attention mechanism is a good way to encode these two types of information. For example, why not add in the temporal information in the node embedding, and then use the spatial information in the attention mechanism. I think the paper could strengthened by adding a discussion about reasons leading to the specific design choices.

2. Lack of comparison and ablation studies: to study the problem above, I feel it would be helpful to conduct more ablation studies investigating the design choices. It is good that the proposed mechanism seems to work well for the two proposed tasks, but I think it would also be worthwhile to see how robust the performance is to the specific design, and how would different components affect the performance (besides the effect of the mask in 4.3).

**Questions:**

1. In equation (2), what is $T_{tt'}$?

2. What are the criteria used when selecting the seizure detection task and the traffic forecasting task in this work? I understand these are important tasks, but given how many potential applications that could involve space-time dynamics, I am wondering what characteristics a task should have that make the proposed framework suitable and effective.

3. Is there a reason why GMask seems only to help with the traffic forecasting task but not the seizure detection task? Could you investigate this further? Also any idea why other masking strategies seem to work worse than even no masking?

---

> ### Author Response · Authors · 2024-11-24
>
> Thank you for your review. We believe that the reviewer may have overlooked some sections of our paper, particularly the ablation study detailed in Appendix K. This could have contributed to the perceived weaknesses and a potential misjudgment of our contributions.
>
> ### Ablation Study on the Choice of Spatio-Temporal Encoding
>
> In light of the weaknesses mentioned in the reviewer's feedback, we would like to emphasize a key ablation component of our paper, which thoroughly investigates the choice of spatio-temporal encoding.
>
> #### **1) Motivation behind the choice of our Spatio-Temporal encoding:**
>
> - **Spatial Encoding**: As highlighted in our main manuscript (lines 240–251), the fixed spatial encoding was chosen because the graph nodes in our dataset and problem setting are fixed in space. This provides a clear motivation for using fixed spatial encoding. Moreover, we complement this fixed spatial information with local graph information via GMask. This combination allows the model to benefit from both the fixed positional information and additional local context, making the fixed encoding a valuable and novel feature for NTAttention.
> - **Temporal Encoding**: For temporal encoding, we selected relative positional encoding based on prior studies that demonstrate its effectiveness for temporal dependencies [1].
>
> #### **2) Ablation Study on the Choice of Encodings Provided in Appendix K**
>
> In alignment with the reviewer's observations, we aimed to ensure that the choice of our spatio-temporal encodings, despite the strong motivation behind them, is indeed effective compared to alternative methods of incorporating such encodings. To address this, we conducted an ablation study evaluating various encoding strategies for seizure detection. These include:
>
> - Fixed Temporal Encoding (Fixed-TE)
> - Rotational Spatial Encoding (Rot-SE)
> - Relative Temporal Encoding (Rel-TE)
> - Fixed Spatial Encoding (Fixed-SE)
>
> This study investigates different combinations of spatial and temporal encodings, as suggested by the reviewer, to validate their performance. Below, we include the corresponding table summarizing the results:
>
>
>
> |**Clip Size**|**Model**|**AUROC**|**F1-Score**|**Sensitivity**|**Specificity**|
> |---------------|------------------------------|------------|--------------|-----------------|-----------------|
> | 12-s          | Fixed-TE + Fixed-SE         | 0.84       | 0.450        | 0.633           | 0.902           |
> || Rot-SE + Rel-TE             | 0.83       | 0.461        | 0.667           | 0.812           |
> || Fixed-TE + Rot-SE           | 0.84       | 0.444        | 0.630           | 0.870           |
> || Fixed-TE + Rot-SE + GMask   | 0.84       | 0.444        | 0.630           | 0.870           |
> || NTAttention + GMask         | 0.827      | 0.434        | 0.612           | **0.922**       |
> || NTAttention                 | **0.842**  | **0.451**    | **0.638**       | 0.904           |
> | 60-s          | Fixed-TE + Fixed-SE         | 0.782      | 0.471        | 0.421           | 0.921           |
> |               | Rot-SE + Rel-TE             | 0.771      | 0.500        | 0.410           | 0.782           |
> |               | Fixed-TE + Rot-SE           | 0.784      | 0.541        | 0.400           | 0.927           |
> |               | Fixed-TE + Rot-SE + GMask   | 0.782      | 0.613        | 0.511           | 0.843           |
> |               | NTAttention + GMask         | 0.791      | 0.475        | 0.410           | 0.920           |
> |               | NTAttention                 | **0.810**  | **0.671**    | **0.489**       | **0.945**       |
>
>
> These results demonstrate that NTAttention's spatio-temporal encodings achieve superior performance compared to all other tested encoding combinations. This validates that our choice of encodings is not only effective but also aligns well with the inherent nature of the data, making it a robust and well-suited approach for the tasks at hand.
>
> **Q1) What is the $T_{tt'}?$**
>
> $T_{tt'}$ Is the reletive temporal encoding between time points $t$ and $t'$.
>
> **Q2) Criteria of choosing the trafic and seizure tasks**
>
> As highlighted in Introduction, traffic forecasting and seizure detection have significant real-world impact on our daily lives. Due to their importance, previous studies have invested considerable effort in addressing these tasks.
>
> We have followed established research in traffic forecasting [4,5] and seizure detection [3,6]. However, these problems have traditionally been studied as separate domains, with models specifically designed to handle one or the other. In contrast, we propose NTAttention, a unified approach capable of solving both tasks effectively.
>
> Furthermore, previous studies have utilized graph-based modeling for these datasets, as they represent temporal graph data in real-world scenarios [4,6]. These two tasks serve as excellent examples of spatiotemporal graph applications that are both impactful and representative of the broader domain.

---

> ### Author Response · Authors · 2024-11-24
>
> **Q3) Effect of GMask in both tasks**
>
> We address the reviewer's question in two parts:
>
> **1) Why is GMask effective in traffic forecasting but less so in seizure detection?**
>
> As mentioned in our manuscript (lines 410–418), GMask is more effective for larger graphs, such as those in the traffic dataset. In such graphs, GMask reduces the influence of connections between distant nodes in the attention mechanism, which can often be considered as noise, as also discussed in prior works like [2]. This filtering enhances the model’s focus on more meaningful, localized interactions, thereby improving performance.
>
> In contrast, for smaller graphs, such as those in the EEG dataset, the interaction between every pair of nodes is crucial due to the limited number of nodes. Applying GMask in this context can lead to a slight reduction in performance. However, GMask still significantly improves computational efficiency, which is critical for scalability (Section 4.4 of our manscript). Further details on the impact of GMask on performance and sparsity are provided in Appendices F and K.
>
> **2) Why other masks are not effective**
>
> As mentioned in Section 4.3 of our manuscript, other masking strategies like BIGBIRD do not account for the specific relationships between graph nodes in NTAttention. These methods randomly mask parts of the attention matrix, which does not align well with the structure of spatiotemporal graphs.
>
> GMask, on the other hand, considers the distance between nodes over all time points. If two nodes are far apart throughout the graph, GMask sets the attention scores between them to zero. This ensures that NTAttention focuses only on meaningful connections while ignoring irrelevant ones. Unlike other masking strategies, GMask is specifically designed for NTAttention, making it more effective for spatiotemporal graph data.
>
> ---
>
> [1] Rethinking and improving
> relative position encoding for vision transformer. In Proceedings of the IEEE/CVF International Conference on Computer Vision, pp. 10033–10041, 2021.
>
> [2] "Semi-supervised classification with graph convolutional networks." arXiv preprint arXiv:1609.02907 (2016).
>
>
>
> [3] Selfsupervised graph neural networks for improved electroencephalographic seizure analysis. arXiv preprint.
>
> [4] Diffusion Convolutional Recurrent Neural
> Network: Data-Driven Traffic Forecasting, February 2018.
>
> [5] Learning to remember patterns: pattern matching memory networks for traffic forecasting
>
>
>
> [6] "Rest: Efficient and accelerated eeg seizure analysis through residual state updates." ICML (2024).

---

> ### Comment · Reviewer_1iQN · 2024-11-25
>
> I thank the authors for their answers. The rebuttal has addressed some of my questions, but there are two answers I find less satisfying:
>
> 1. Why such a design choice: while I appreciate the ablation studies and explanation, I think there is still one question the paper should answer: while it is clearer to me now why the spatial and temporal encodings are individually chosen in the current way, my original question was to also ask about why they are combined in such a way. For example, why not add in the temporal information in the node embedding, and then use the spatial information in the attention mechanism? I think it would be a good addition if the work can also provide more insights into this overall design.
>
> 2. Criteria of choosing the traffic and seizure tasks: like I said, I understand that these are important tasks, but there are many other tasks that also have a spatiotemporal graph representation: for example, multi-agent communication. My question is that what characteristics should a spatiotemporal task have so that the method would be appropriate and effective? It would be nice if the paper could provide more insights into the robustness and generalizable aspects of the method.
>
>
> Moreover, I feel there are several insightful points in the rebuttal that are currently missing in the paper, such as why is GMask effective in traffic forecasting but less so in seizure detection and why GMask has such a design. I will increase my score to 5 and encourage the authors to include more of their insights into the paper.

---

> > ### Author Response · Authors · 2024-11-28
> >
> > We thank the reviewer for increasing the score and encuraging us to add the discussion in the rebuttal in the main body of the paper. We would like to mention that **we have revised the paper manscript based on the reviewers suggestions and in our revised pdf the new material added from the rebuttal is highlighted in blue text**.
> >
> > We have added a discussion explaining why GMask is more effective in traffic forecasting compared to seizure detection, as well as the rationale behind this design choice. Below, we address your remaining concerns in detail:
> >
> > ---
> >
> > **1)Why such a design choice**
> >
> > We would like to clarify that adding temporal encoding to the nodes corresponds to the Fixed-TE discussed in Appendix Section K. Since relative temporal encoding cannot be added independently to the node features, our ablation of Fixed-TE specifically refers to adding temporal encodings to the graph nodes, which has already been covered in our ablation study.
> >
> > Similarly, for spatial encoding, when we applied rotational spatial encoding, it was incorporated into the queries and keys as learnable vectors, analogous to how our current temporal encoding is implemented. These experiments and analyses are already included in Appendix K, where we provide a comprehensive evaluation of these encoding strategies.
> >
> > Additionally, regarding the insights, as mentioned earlier, the datasets in our study consist of graphs with fixed nodes. This naturally motivates the addition of fixed (non-rotational) spatial encoding for nodes, as it aligns with the inherent structure of the data.
> >
> > The motivation for using rotational temporal encoding, on the other hand, is based on findings from previous studies such as [1], which highlight its effectiveness for capturing temporal dependencies. Our ablation study further supports this choice, demonstrating the suitability of fixed spatial encoding and rotational temporal encoding for the tasks at hand.
> >
> > ---
> >
> > **2)Criteria of choosing the traffic and seizure tasks**
> >
> > Thank you for the question. The main motivation behind our work was to develop a simple yet effective and understandable transformer model specifically designed for datasets with fixed sensors collecting time-series data over time. The two tasks we focused on—traffic forecasting and seizure detection—are both real-world, impactful applications that align well with the scope of this study.
> >
> > Previous literature in top ML conferences has primarily focused on either traffic forecasting [2,3,6] or seizure detection [4,5] as separate tasks. For instance, the DCRNN model was originally applied to traffic forecasting in 2018 at ICLR [2], and an adaptation of the same model was later applied to seizure detection at ICLR 2022 [4]. Observing this separation in prior studies, we were motivated to design a unified, efficient, and high-performing model that bridges this gap and performs effectively on both types of datasets. We believe this contribution fills an important gap in the current literature.
> >
> > While our study focuses on these two applications, we are open to extending it to other domains, including the example suggested by the reviewer. However, the primary reason for selecting these tasks was the significant gap we identified in existing work. To clarify this motivation further, we have added it explicitly to our paper [153-157].
> >
> >
> >
> > ---
> >
> > **References**
> >
> > [1] Kan Wu, Houwen Peng, Minghao Chen, Jianlong Fu, and Hongyang Chao. Rethinking and improving
> > relative position encoding for vision transformer. In Proceedings of the IEEE/CVF International
> > Conference on Computer Vision, pp. 10033–10041, 2021.
> >
> > [2] Diffusion Convolutional Recurrent Neural Network: Data-Driven Traffic Forecasting, February 2018.
> >
> > [3] Learning to remember patterns: pattern matching memory networks for traffic forecasting
> >
> > [4] Selfsupervised graph neural networks for improved electroencephalographic seizure analysis. arXiv preprint.
> >
> > [5] "Rest: Efficient and accelerated eeg seizure analysis through residual state updates." ICML (2024).
> >
> > [6] “Adaptive graph convolutional recurrent network for traffic forecasting”, NeurIPS 2020
> >
> > ---
> >
> > Thank you, Reviewer, for your thoughtful feedback and engagement. We have addressed the concerns raised and provided detailed responses to clarify all points. We kindly request you to reconsider your score in light of the explanations and updates we have shared. If there are any additional concerns or questions, we are happy to address them further.
> >
> > Thank you for your time and consideration!

---

### Official Review · Reviewer_4UxP · 2024-11-03

**Soundness:** 2
**Presentation:** 2
**Contribution:** 1
**Rating:** 3
**Confidence:** 4

**Summary:**

This study proposed NTAttention for targeting seizure detection and traffic forecasting.NTAttention effectively utilizes the inherent positional and temporal information in the data by integrating spatial encoding into the features of each node and integrating relative temporal encoding into the attention matrix. Simultaneously using geometric perceptual masks to improve model efficiency.

**Strengths:**

1.The authors address the challenge of capturing long-range dependencies in temporal graph signals, which is a non-trivial extension of existing methods.
2.By integrating spatial encoding into node features and relative temporal encoding into the attention matrix, NTAttention combines graph topology with temporal dynamics in a creative way.
3. The authors provide a solid theoretical basis for NTAttention, including the formulation of the attention mechanism and the geometry-aware masking (GMask).

**Weaknesses:**

1.The authors demonstrate the effectiveness of NTAttention on EEG seizure detection and traffic forecasting. However,
testing the model on more diverse datasets, particularly those with different characteristics, could further validate the
generalizability of the approach.
2.This article compares NTAttention with several baselines, but can be compared with the latest developments in GNN models,
especially those designed for spatiotemporal data, which can provide more rigorous benchmarks.
3.Providing theoretical insights into the convergence properties of NTAttention, especially in the context of non-stationary graph signals, could be a valuable contribution

**Questions:**

1. Generalizability Across Datasets:
Will the authors consider testing NTAttention on additional datasets to further demonstrate its generalizability beyond
EEG seizure detection and traffic forecasting?
2. Comparison with State-of-the-Art GNNs:
Could the authors comment on why they chose not to compare NTAttention with the most recent GNN models designed for spatio-temporal data, and would they consider adding these comparisons in a revised version?
3. Data Privacy and Ethical Considerations:
Could the authors discuss how they ensure data privacy, particularly for sensitive health data used in the EEG seizure
detection task?
4. Scalability to Large Graphs:
Can the authors provide more details on the scalability of NTAttention to very large graphs, including any potential limitations they have encountered?

---

> ### Author Response · Authors · 2024-11-24
>
> **W1, Q1) Generalizability Across Datasets**
>
> Unlike prior studies that focused on a single dataset for a specific task—such as DCRNN [1] applied only to seizure detection on the TUSZ dataset or PM-MemNet [3] used exclusively for traffic forecasting—NTAttention is the first model to be applied to both tasks using multiple real-world datasets. This highlights its versatility and generalizability across diverse applications.
>
> Regarding the datasets, the TUSZ dataset, which includes recordings from 579 patients, is considered large and comprehensive within the seizure detection community. It is widely recognized as valid and sufficient for benchmarking seizure detection models [1, 4]. Additionally, the datasets in our study encompass both large graphs (for traffic forecasting) and small graphs (for EEG-based seizure detection), ensuring that NTAttention is thoroughly tested on a variety of graph scales.
>
> As mentioned in the introduction, NTAttention is specifically designed to handle fixed-node graphs with temporal signals. The two tasks we address—traffic forecasting and seizure detection—are exemplary use cases for this type of data, as they have been extensively studied in previous research. These tasks provide an ideal benchmark to demonstrate the effectiveness of NTAttention in handling spatiotemporal dependencies in graph-structured data.
>
> The selection of tasks in our study is based on well-established research in these fields, as highlighted in studies such as [1, 2, 3, 4, 5], which have specifically addressed these tasks. These applications are not only well-researched but also highly impactful in real-world scenarios, playing a crucial role in our daily lives. While prior studies have typically focused on only one of these applications, NTAttention has demonstrated its effectiveness across both, highlighting its versatility.
>
> The datasets chosen for spatiotemporal graph forecasting are also widely recognized and commonly used in this domain. However, if the reviewer has suggestions for other applications or datasets, we would be grateful for the recommendation and would be happy to evaluate NTAttention on them.
>
> We believe these points address concerns about dataset sufficiency and demonstrate the broad applicability of our model.
>
> **W2, Q2) Comparison with State-of-the-Art GNNs**
>
> We would like to highlight that we benchmarked NTAttention against well-known, recent, and highly relevant works specifically designed for real-world tasks. For example, REST [5] (2024) and STAEFormer [6] (2023) are state-of-the-art models in the context and scope of our study. These models are not only cutting-edge but are also tailored to address the complexities of real-world problems such as traffic forecasting and seizure detection.
>
> **Q3) Data Privacy and Ethical Consideration**
>
> The EEG Seizure Corpus from Temple University Hospital, utilized in our research, is anonymized and publicly accessible with IRB approval [7]. This dataset is openly available for download at [TUSZ](https://isip.piconepress.com/projects/tuh_eeg/), and has been widely used in previous benchmarks such as [4, 1, 5]. Additionally, the privacy and usage policies for this dataset are clearly outlined at [Policy](https://isip.piconepress.com/projects/nedc/html/tuh_eeg/).
>
> **W3 & Q4) Scalability to Large Graphs**
>
> For spatiotemporal graphs, the number of nodes typically does not exceed a few hundred, and NTAttention is specifically designed to handle tasks of this scale, as supported by prior studies [1, 3, 5]. Additionally, we have analyzed the efficiency of NTAttention with and without GMask in Appendix G, which demonstrates the scalability of GMask and the overall model. Furthermore, a theoretical complexity analysis is provided in Appendix J of the paper, where we discuss how sparsification becomes increasingly effective for larger graphs since more nodes are distant (and can be filtered based on a threshold $k$).
>
> We refer to Appendices G and J for a more detailed discussion on the efficiency, scalability, and theoretical analysis of NTAttention and GMask.
>
> [1] Selfsupervised graph neural networks for improved electroencephalographic seizure analysis. arXiv preprint.
> [2] Diffusion Convolutional Recurrent Neural Network: Data-Driven Traffic Forecasting, February 2018.
> [3] Learning to remember patterns: pattern matching memory networks for traffic forecasting. ICLR 2022
> [4] Graph-generative neural network for EEG-based epileptic seizure detection via discovery of dynamic brain functional connectivity.
> [5] Rest: Efficient and accelerated eeg seizure analysis through residual state updates. ICML (2024).
> [6] Spatio-temporal adaptive embedding makes vanilla transformer sota for traffic forecasting. In Proceedings of the 32nd ACM international conference on information and knowledge management, pp. 4125–4129, 2023.
> [7] The temple university hospital eeg data corpus. Frontiers in neuroscience, 10:196, 2016.

---

> ### Author Response · Authors · 2024-11-28
>
> We would like to gently remind the reviewer that the deadline of the author-reviewer discussion period is approaching.
>
> We appreciate your valuable contributions. We believe that our responses address the concerns raised. If you have any further suggestions or concerns, please let us know.
>
> Best regards,
>
> The Authors

---

> > ### Author Response · Authors · 2024-12-02
> > **Final Day of Discussion Period for ICLR 2025**
> >
> > ---
> >
> > Dear Reviewer 4UxP,
> >
> > We hope you are doing well. As today **marks the last day of the extended discussion period for ICLR 2025**, we would like to kindly request your final feedback on our submission. We sincerely appreciate your thoughtful comments, and we have made every effort to address your concerns in our rebuttal.
> >
> > If there are any remaining questions or clarifications needed, we would be more than happy to respond. **We would greatly appreciate it if you could review the rebuttal and provide your final thoughts by the end of the discussion period.**
> >
> > Additionally, if you feel that we have adequately addressed all concerns, **we would greatly appreciate it if you could consider increasing the score accordingly.**
> >
> > Best regards,
> >
> > Authors
> >
> > ---

---

### Official Review · Reviewer_tACj · 2024-11-04

**Soundness:** 2
**Presentation:** 1
**Contribution:** 1
**Rating:** 1
**Confidence:** 4

**Summary:**

- This paper introduces NTAttention, a novel self-attention mechanism designed for processing temporal graph signals. The proposed method allows nodes to attend to both spatial and temporal dimensions simultaneously while incorporating graph topology through geometry-aware masking (GMask).
- The authors show the effectiveness of their approach on 2 applications: EEG-based seizure detection and traffic forecasting.
- They achieve state-of-the-art performance on 2 applications.
- The proposed method maintains computational efficiency through GMask while offering a flexible framework.

**Strengths:**

- The authors introduce an innovative unified attention mechanism that simultaneously handles node and temporal relations.
- The proposed method achieves state-of-the-art performance in two distinct applications.
- The proposed method demonstrates robustness across multiple domains with minimal architectural changes.
- The proposed method improves performance while maintaining computational efficiency, making it practical for real-world deployments.
- The geometry-aware masking strategy provides a general approach that may benefit other attention-based architectures.

**Weaknesses:**

- The paper's primary contribution appears incremental rather than novel:
  - The NTAttention mechanism is essentially a standard Transformer with added spatio-temporal encoding
  - The primary motivation of capturing "long-range dependencies" conflicts with GMask's function of blocking distant node attention
  - The complexity analysis $(O(N^2T^2))$ contradicts claims of computational efficiency
  - The theoretical foundation for selecting GMask threshold $k$ appears heuristic.

- The experimental evaluation lacks crucial analyses.
  - No rigorous comparison between GMask and random masking with equivalent masking ratios
  - Missing analysis of attention weight distributions compared to baseline models
  - Absence of ablation studies on the impact of different masking strategies in seizure detection
  - Limited scale validation (only 19 and 207 nodes)

- Several technical aspects require clarification.
  - The motivation for 2D coordinate-based spatial encoding versus traditional graph topology representations
  - The relationship between the attention mechanism and traditional GNN message passing
  - The stability guarantees for the learning process, especially with small k values in GMask
  - The impact of GMask on optimization dynamics

- Table 1's binary comparison (checkmark/x-mark) oversimplifies complex methodological differences
- The paper's title "Graph Signal Processing" is misleading.

**Questions:**

**Visualization and Presentation Issues:**
1. Why are $v_i^t$ and $v_j^{t'}$ depicted at position (5,5) on the diagonal in the self-attention matrix in Figure 1? This visualization needs clarification or correction.

2. The related work section on Traffic Forecasting primarily references studies up to 2021. Please include more recent developments in this rapidly evolving field.

**Comparative Analysis Concerns:**

3. Table 1 appears overstated. Notably, REST [1] shows a checkmark for criterion "C" in their paper, yet your table marks it with an x-mark. Please explain this discrepancy.

4. The binary (checkmark/x-mark) representation in Table 1 oversimplifies the comparison. A more thorough analysis of similarities and differences would serve readers better and avoid potential misunderstandings.

**Masking Analysis:**

5. Table 7's comparison between GMask and RandomMask requires more rigorous analysis. Can the authors compare where the random masking ratio matches GMask's masking ratio?

6. Please explain why "No Mask" performs comparable to GMask while RandomMask, WindowMask, and BigBird show significantly higher errors.

7. Can the authors extend the analysis in Table 7 to include Seizure Detection results?

**Methodological Clarifications:**

8. How does NTAttention differentiate itself from existing Transformer-based time-series forecasting models beyond adding spatio-temporal encoding?

9. The selection of GMask's threshold parameter $k$ appears heuristic. Can the authors provide domain-specific insights into why optimal $k$ values differ between traffic forecasting and seizure detection?

10. Particularly, why does your method perform best with $k=0$ for seizure detection when REST[1] reports optimal performance at k=0.9?

11. Given that STGCN[2] uses a similar adjacency matrix creation method with a threshold of 0.5, does this suggest similar operational principles between masking matrices and graph convolution in traffic forecasting?

**Performance Analysis:**

12. Please analyze why the GMask application degrades performance in seizure detection (Table 5).

13. The paper's emphasis on capturing "long-range dependencies" seems to contradict GMask's function of blocking attention between distant nodes. Can the authors quantify the extent of this attention blocking to better support your claims?

**Comparative Framework:**

14. In traffic forecasting, Please include and compare with recent studies using adaptive adjacency matrices[3,4] and alternative spatial dependency approaches[5,6].

15. Can the authors analyze the distributional differences between your attention matrices and the adjacency matrices used in STGCN and DCRNN?

16. How do your attention weights compare with other self-attention baseline models?

**Theoretical Concerns:**

17. The claim that "traditional methods have struggled to account for long-range dependencies" needs stronger experimental or theoretical support.

18. Please provide detailed analysis of how $\alpha(N)$ affects computational complexity in practical applications, given the $\mathcal{O}(N^2T^2)$ theoretical complexity.

19. Is 2D coordinate encoding sufficient for representing graph topology? How does this compare to Hadamard Attention's use of pre-defined adjacency?

20. Can the authors discuss the expressiveness of your attention mechanism compared to traditional GNN message passing?

21. How do you ensure learning stability with GMask, particularly regarding potential node isolation with small $k$ values?

22. From the perspective of model optimization, the proposed structure of GMask seems to affect the optimization of the model. Can you discuss this further?

**Title Recommendation:**

23. The title "Graph Signal Processing" may mislead readers as the paper doesn't employ traditional GSP concepts (graph Fourier transform, spectral analysis, graph wavelets). Consider revising to reflect the actual content better.

> [1] Afzal, Arshia, et al. "Rest: Efficient and accelerated eeg seizure analysis through residual state updates." ICML 2024
>
> [2] Yu, Bing, Haoteng Yin, and Zhanxing Zhu. "Spatio-temporal graph convolutional networks: a deep learning framework for traffic forecasting." IJCAI 2018.
>
> [3] Bai, Lei, et al. "Adaptive graph convolutional recurrent network for traffic forecasting.” NeurIPS 2020
>
> [4] Choi, Jeongwhan, et al. "Graph neural controlled differential equations for traffic forecasting." AAAI 2022
>
> [5] Fang, Zheng, et al. "Spatial-temporal graph ode networks for traffic flow forecasting." KDD 2021


---


---

# $\color{red}{\text{Final Recommendation to Area Chair}}$ **(Edited at 2:15 PM on 3rd Dec (AOE))**

During the final stage of discussion with the authors, I discovered a critical flaw that drive me to reduce my score from `3` to `1`. Through careful code inspection, I identified that their unusual error trends in different forecasting horizons stemmed from a fundamentally incorrect evaluation protocol (see `Table 6`).

The authors' implementation reveals they train separate models for different forecasting horizons (3, 6, 12 steps) by using only the last few timestamps of historical data, as evidenced by their code:

```python
self.timepoints = 3
self.node_time_proj = torch.nn.Linear(207*self.timepoints, 207)
```

This approach fundamentally violates the standard practice in traffic forecasting where:
1) models receive a complete 12-step historical sequence as input
2) forecast a complete 12-step future sequence
3) report metrics for horizons 3, 6, and 12 from this single predicted sequence

The authors' claim of fair comparison is weakened by this serious error, which makes their results incomparable with existing work and could mislead future researchers in the spatiotemporal forecasting community.

The authors need to completely revise their experimental setup to align with established practices in traffic forecasting. Thus, I believe the reduction in score is warranted.

---

> ### Author Response · Authors · 2024-11-24
>
> Thank you for the reviewer's detailed response and comments. We greatly appreciate the feedback and will address each point below in order:
>
> **Q1) Why are $v^t_i$ and $v^{t'}_j$ in the diagonal part of attention in figure1**
>
> Thanks for the pointer, we will revise Figure 1 in the final version.
>
> ---
>
> **Q2) Related work section for traffic forecasting can include more recent studies**
>
> Thank you for the comment. We have already included 13 baselines for benchmarking, including several recent models published as recently as 2023, such as PM-MemNet (ICLR 2022). If there are additional baselines that you have in mind, we would be happy to include them in our study.
>
> ---
>
> **Q3) Table 1 appears overstated. REST has checkmark for item C**
>
> The REST model has been evaluated on window sizes of up to 14 seconds, which is relatively short compared to models like DCRNN [1], which evaluate seizure detection on windows as long as 1 minute, often referred to as long-term detection. Due to this difference, we believe the evaluation window size of the REST model is not sufficient for capturing long-term dependencies compared to state-of-the-art approaches. In contrast, both NTAttention and DCRNN have been evaluated on significantly larger window sizes, up to 1 minute, which is five times longer than the window size used by REST. This allows NTAttention and DCRNN to better capture long-term patterns in seizure detection.
>
> ---
>
> **Q4) The binary (checkmark/x-mark) representation in Table 1 oversimplifies the comparison**
>
> Thank you for the comment. We have already included more details about related work in Appendix Section C, providing an in-depth explanation of how each model is built. Additionally, our numerical evaluations comprehensively demonstrate the capabilities and performance of each model, highlighting their strengths and comparative effectiveness.
>
> ---
>
> **Q5) Compare where the random masking ratio matches GMask's masking ratio**
>
> Thanks for the comment. We have matched the ratio for binary mask and GMask and results are below on the traffic dataset:
>
> **Table tACj-1**: GMask vs random performance.
> |Mask|$k=0.1$|$k=0.5$|$k=0.9$|
> |-|-|-|-|
> GMask|**5.45**|**4.34**|**5.21**|
> Random|17.2|14.5|12.22|
>
> The results show that GMask is performant compared to random mask regardless of treshold $k$.
>
> ---
>
> **Q6) Comparison between GMask and other type of Masks**
>
> As explained in Section 4.3 of our manuscript, masking strategies such as BIGBIRD fail to account for the unique relationships between graph nodes in NTAttention. These approaches apply random masking to parts of the attention matrix, which does not effectively capture the structure of spatiotemporal graphs.
>
> In contrast, GMask is designed to account for the distance between nodes across all time points. If two nodes remain far apart throughout the graph, GMask masks their attention scores by setting them to zero. This allows NTAttention to focus on relevant connections while filtering out less meaningful ones. Unlike other masking methods, GMask is tailored specifically for NTAttention, making it highly effective for spatiotemporal graph data.
>
> ---
>
> **Q7) Expanding Table 7 with seizure detection results**
>
> Thanks for the pointer, we will add the results of seizure detection to Table 7.
>
> ---
>
>
> **Q8) How does NTAttention differs to exisiting models**
>
> While several models, such as REST [2] and DCRNN [1], leverage graph convolutional operations, NTAttention distinguishes itself from other attention-based graph models. Unlike approaches such as STAFormer [3], NTAttention applies attention mechanisms across all temporal points and between all nodes within the graph simultaneously.
>
> ---
>
> **Q9) Selection of $k$ in GMask**
>
> The choice of the parameter $k$ is dependent on the specific task and application. Additionally, a key consideration is that the choice of $k$ is influenced by the number of nodes in the graph. A clear example of this can be seen in the DCRNN model, which uses different values of $k$ and graph structures for traffic forecasting [4, ICLR 2018] and seizure detection [1, ICLR 2022]. This illustrates how $k$ must be tailored to the characteristics of the task and the underlying graph.
>
> ---
>
> **Q10) Why does NTAttention uses $k=0$ and REST uses $k=0.9$?**
>
> As mentioned in our response to the previous question, the parameter $k$ is a hyperparameter that is tuned based on the specific model and task. In REST, the adjacency matrix defined by $k$ is used for graph convolution (GConv), whereas in NTAttention, it is employed for masking the attention scores. These are fundamentally different architectural approaches, which is why the parameter $k$ is tuned differently for different baselines, reflecting the distinct roles it plays in each model.

---

> ### Author Response · Authors · 2024-11-24
>
> **Q11) STGCN uses similar adjacency matrix**
>
> Different architectures recommend varying values for the parameter, such as REST using 0.9 and STGCN using 0.5, due to differences in their underlying designs. However, there might be interesting connections between the parameter choices for GConv-based models like STGCN and attention-based models like NTAttention. This is a valuable insight raised by the reviewer and could provide an interesting direction for future research to explore potential relationships between these approaches.
>
> ---
>
> **Q12) GMask application degrades performance in seizure detection**
>
> As mentioned in Section 4.1 of our paper, we believe this is due to the small number of nodes. In scenarios with fewer nodes, every individual interaction becomes crucial, and applying GMask may lead to increased efficiency but with a slight degradation in performance. However, for traffic forecasting, which involves a large number of nodes (270 in our case), considering all interactions between every pair of nodes introduces noise and redundant information, particularly between distant nodes, as noted in prior studies [5]. GMask effectively mitigates this issue by focusing attention on more relevant interactions.
>
> ---
>
> **Q13) Long-range dependencies in contradiction with GMask**
>
> We would like to emphasize that GMask only masks interactions between distant nodes, while close nodes (those not masked by GMask) still attend to all time points of each other. This approach is not contradictory to the concept of capturing long-range dependencies. Similar to transformers, where masking can enhance both efficiency and accuracy, we observe that masking certain parts of the attention matrix with GMask can lead to improved accuracy while simultaneously increasing computational efficiency.
>
> ---
>
> **Q15) Distributional differences between your attention matrices and the adjacency matrices used in STGCN and DCRNN**
>
> The attention matrix in NTAttention, STGCN, and REST all follow the standard distance-based adjacency matrix and share the same fundamental formulation. However, NTAttention applies this adjacency matrix as a mask for its attention mechanism, whereas in STGCN and REST, it is utilized for graph convolution. This key difference reflects how each model leverages the adjacency matrix to suit its architecture and task requirements.
>
> ---
>
> **Q16) How do your attention weights compare with other self-attention baseline models?**
>
> All previous benchmarks utilize separate attention mechanisms for space and time, NTAttention instead applies a unified attention mechanism across both space and time, resulting in an attention matrix of size $NT \times NT$. Due to this fundamental difference in the strategy for constructing the attention, the exact weights are not directly comparable between NTAttention and other models.
>
> ---
>
> **Q17) "Traditional methods have struggled to account for long-range dependencies" needs stronger experimental or theoretical support.**
>
> Thank you for the pointer. This has also been discussed in previous studies, such as REST [2], GMAN [6], and STAEFormer [3]. We will add these citations to the statement for stronger support and to provide better context.
>
> ---
>
> **Q18) Details of how $\alpha(N)$ affects computational complexity**
>
> Thanks for the pointer we would like to highlight that we already have analyzed the effect of the threshold $k$ in our appendix section G which shows how exactly the parameter $k$ affects the number of FLOPS and also we have added the results on the effect of parameter $k$ on the accuracy at table in our response Q5.
>
> ---
>
> **Q19) Is 2D coordinate encoding sufficient for representing graph topology?**
>
> Thank you for the comment. For the case of EEG, we have utilized the exact 3D distances between nodes, which are also available in the DCRNN repository at [GITHUB](https://github.com/tsy935/eeg-gnn-ssl/blob/main/data/electrode_graph/distances_3d.csv).
>
> For traffic forecasting, since the z-coordinate (altitude) of the sensor locations does not vary significantly compared to the x and y coordinates, we believe a 2D representation is sufficient. Moreover, the dataset provides only x and y coordinates, and we have built the graph accordingly. This ensures the graph construction aligns with the data's spatial resolution.

---

> ### Author Response · Authors · 2024-11-24
>
> **Q20) The expressiveness of your attention mechanism compared to traditional GNN message passing?**
>
> Traditional convolutional message passing is less expressive than attentional mechanism.
> Indeed, attentional message passing can represent convolutional by an attention mechanism implemented as a look-up table with attention weight set as the edge weight $A_{n,n'} = c_{n,n'}$ at fixed time $t$, where $c_{n,n'}$ is the edge weight between node $n$ and node $n'$ [7, page 79].
> Instead of fixed, we learn this weight learned through the NTAttention mechanism.
>
> ---
>
> **Q21) Stability with GMask**
>
> GMask sets the attention of spatially distant nodes to exactly 0 for exploit efficient sparse matrix multiplication routines.
> The value of $k$ can be regarded as a sparsity parameter, where training stability is preserved even at high sparsity percentages as shown in Table tACj-1, where the best result is obtained with the lowest $k$.
>
>
> ---
>
> **Q22) Structure of GMask seems to affect the optimization of the model. Can you discuss this further?**
>
> GMask improves model optimization it by inducing sparsity in the attention matrix. This sparsity reduces the number of computations, effectively acting as a regularizer, similar to dropout. Furthermore, the structured sparsity induced by GMask aligns with the graph topology, focusing attention on relevant nodes and further enhancing optimization towards a solution that respects the underlying graph structure. A detailed theoretical analysis of the impact of GMask on computational complexity is available in Appendix J. Empirically, the faster training times with GMask (Table 8 and Table 10) corroborate the theoretical analysis and demonstrate improved optimization efficiency.
>
>
>
> ---
>
> **Q23) The title "Graph Signal Processing" may mislead readers**
>
> Thanks for your suggestion. We will change the title to: "EEG seizure detection and traffic forecasting with space-time self-attention"
> to match better our contributions and work.
>
> ---
>
>
> [1] Selfsupervised graph neural networks for improved electroencephalographic seizure analysis. arXiv preprint.
>
> [2] "Rest: Efficient and accelerated eeg seizure analysis through residual state updates." ICML (2024).
>
> [3] Spatio-temporal adaptive embedding makes vanilla transformer sota for traffic forecasting. In Proceedings of the 32nd ACM international conference on information and knowledge management, pp. 4125–4129, 2023
>
> [4] Diffusion Convolutional Recurrent Neural
> Network: Data-Driven Traffic Forecasting, February 2018.
>
>
> [5] "Semi-supervised classification with graph convolutional networks." arXiv preprint arXiv:1609.02907 (2016).
>
>
> [6] Gman: A graph multi-attention network for traffic prediction. In Proceedings of the AAAI conference on artificial intelligence,
> volume 34, pp. 1234–1241, 2020.
>
> [7] "Geometric deep learning: Grids, groups, graphs, geodesics, and gauges." arXiv preprint arXiv:2104.13478 (2021).

---

> ### Comment · Reviewer_tACj · 2024-11-25
>
> Thank you for your responses. While most of your answers address my questions and concerns, I still have remaining questions and suggestion:
>
> ---
>
> **Suggestion.** Rather than stating, "We will revise Figure 1 in the final version," I recommend providing an updated PDF with tracked changes. It would be helpful to see the specific modifications clearly marked in an updated document.
>
> **Q1.** You state that NTAttention and DCRNN better capture long-term patterns in seizure detection. Does this mean you plan to modify Table 1? There seems to be a discrepancy between your response and the table's current representation that needs clarification.
>
> **Q2.** The citation of DCRNN[2] in the seizure detection context appears inappropriate. I suggest citing the original DCRNN paper [2] for spatio-temporal forecasting and clearly distinguishing Dist-DCRNN/Corr-DCRNN from [1]. Note that while Section 4.2 correctly cites [2], Table 2 incorrectly references [1].
>
> **Q3.** Why weren't the models "Corr-DCRNN w/ Pre-training" and "Dist-DCRNN w/ Pre-training" from [1] included in your comparison? A discussion of results against these original models seems necessary.
>
> **Q4.** Could you explain why you didn't use the Weighted F1-Score as reported in [1] for Table 5?
>
> **Q5.** Your response defers the comparison between STGCN's adjacency matrix and NTAttention to future research. However, this discussion period would be an appropriate time to clarify these differences and articulate NTAttention's true novelty.
>
> **Q6.** Your response to previous Q12 cites [3] to support claims about noise and redundant information in traffic forecasting with many nodes. However, [3] primarily discusses node classification tasks and GCN models, not traffic forecasting. Furthermore, follow-up research suggests that adaptive adjacency matrices[4,5] without masking outperform approaches (e.g., STGCN's adjacency matrix) similar to your GMask mechanism. Could you clarify your position on these findings and address how you plan to strengthen your paper's arguments? Note that respond to the related Q14, which was not addressed in your responses
>
> > [1] "Self-supervised graph neural networks for improved electroencephalographic seizure analysis”, ICLR 2022.
> >
> > [2] [“Diffusion Convolutional Recurrent Neural Network: Data-Driven Traffic Forecasting”, ICLR 2018.](https://openreview.net/forum?id=SJiHXGWAZ)
> >
> > [3] Semi-supervised classification with graph convolutional networks”, ICLR 2017.
> >
> > [4] “Adaptive graph convolutional recurrent network for traffic forecasting”, NeurIPS 2020
> >
> > [5] "Graph neural controlled differential equations for traffic forecasting”, AAAI 2022

---

> > ### Author Response · Authors · 2024-11-28
> > **Authors Rebuttal (1/2)**
> >
> > Thanks for the suggestion **we have uploaded the new version of the manscript with the parts that are changed in blue text**. And we revised the comments and suggestions raised by you and other reviewers in the main text and appendix.
> >
> > **Q1) Modifying Table 1 for DCRNN**
> >
> > Thanks for the pointer we have updated the Table 1 of our paper.
> >
> > ---
> >
> > **Q2) Citation of DCRNN paper in Table 2**
> >
> > Thank you for bringing this to our attention. The citation for Table 2 was indeed incorrect, and we have updated it to include the proper citation for the DCRNN paper regarding traffic forecasting. We appreciate your careful review and feedback!
> >
> > ---
> >
> > **Q3) Why weren't the models "Corr-DCRNN w/ Pre-training" and "Dist-DCRNN w/ Pre-training" included in your comparison?**
> >
> >
> > As their names suggest, the models "Corr-DCRNN w/ Pre-training" and "Dist-DCRNN w/ Pre-training" **utilize pretraining on EEG signals**, whereas all other existing benchmarks do not employ pretraining. Consequently, it would **be unfair to directly compare these baselines with other benchmarks**, as pretraining provides a significant advantage. To ensure a more fair comparison, we included the original versions of these models, "Dist-DCRNN" and "Corr-DCRNN," without pretraining, alongside the other benchmarks.
> >
> > ---
> >
> > **Q4) Could you explain why you didn't use the Weighted F1-Score as reported in [1] for Table 5?**
> >
> > Thanks for your pointer, we have indeed **used the weighted F1-score for the seizure classification task**. To improve clarity, we have updated the metric name from **"F1-Score" to "Weighted F1-Score" in Table 5.**
> >
> > ---
> >
> > **Q5)  Your response defers the comparison between STGCN's adjacency matrix and NTAttention to future research. However, this discussion period would be an appropriate time to clarify these differences and articulate NTAttention's true novelty.**
> >
> > We would like to thank the reviewer for the comment and clarify that our goal was not to defer the comparison to future work. Both GMask in NTAttention and the adjacency matrix in the STGCN model are based on the same Gaussian-thresholded adjacency matrix, as described in [2] and also used in the Dist-DCRNN model [1].
> >
> > The key difference lies in how this adjacency matrix is utilized: STGCN employs it for graph convolution, whereas NTAttention uses it to mask attention scores. Specifically, for two nodes where the attention is not masked, they attend to all time points within each other's windows.
> >
> > The method for creating distance-based adjacency matrices for graphs is well-established in the GNN literature, and we adopt the same approach for GMask. However, the novelty lies in how NTAttention applies this adjacency matrix as a mask in the attention mechanism, as opposed to using it for graph convolution like in STGCN. We believe this is a significant contribution and highlights the uniqueness of NTAttention.
> >
> > ---
> >
> > **Q6. Your response to previous Q12 cites [3] to support claims about noise and redundant information in traffic forecasting with many nodes. However, [3] primarily discusses node classification tasks and GCN models, not traffic forecasting. Furthermore, follow-up research suggests that adaptive adjacency matrices[4,5] without masking outperform approaches (e.g., STGCN's adjacency matrix) similar to your GMask mechanism. Could you clarify your position on these findings and address how you plan to strengthen your paper's arguments? Note that respond to the related Q14, which was not addressed in your responses**
> >
> >
> > Thank you for the comment. First, we would like to direct the reviewer’s attention to **Appendix M, where we have tested GMask with a dynamic (or adaptive) adjacency matrix**. Our experiments showed that this approach underperformed compared to our choice of a distance-based adjacency matrix for GMask.
> >
> > Secondly, as highlighted in [4, 5], the models use RNNs for capturing temporal dynamics and employ an adaptive adjacency matrix for spatial dynamics. However, NTAttention **inherently incorporates this adaptivity through its attention mechanism**. Specifically, in [4], the adjacency matrix is learned using an attention mechanism, which is a feature NTAttention already integrates natively.
> >
> > In NTAttention, the role of GMask is to **explicitly inject relative distances between nodes based on topological information**. This complements the adaptivity of the attention mechanism and has shown measurable improvements in performance. This hybrid approach leverages both the dynamic adaptivity of attention and the structural information provided by GMask, demonstrating its effectiveness.

---

> > > ### Author Response · Authors · 2024-11-28
> > > **Authors Rebuttal (2/2)**
> > >
> > > **Q14) In traffic forecasting, Please include and compare with recent studies using adaptive adjacency matrices[3,4] and alternative spatial dependency approaches[5,6].**
> > >
> > > Thank you for the reminder, and we apologize for not including an answer to this question in our initial response. We would like to point out that reference [6] mentioned in your question is not defined in your provided references, which currently end at number 5. If you could provide the details for this reference, we would greatly appreciate it.
> > >
> > > Regarding approaches [3,4], we are running the currently experiments for them on the METR-LA dataset, as these methods have only been evaluated on the PeMSD4 dataset so far. Thank you for bringing this to our attention.
> > >
> > >
> > > ---
> > >
> > >
> > > **Refrences**
> > >
> > > [1] "Self-supervised graph neural networks for improved electroencephalographic seizure analysis”, ICLR 2022.
> > >
> > > [2] David I Shuman, Sunil K Narang, Pascal Frossard, Antonio Ortega, and Pierre Vandergheynst.
> > > The emerging field of signal processing on graphs: Extending high-dimensional data analysis to
> > > networks and other irregular domains. IEEE signal processing magazine, 30(3):83–98, 2013.
> > >
> > > [3] Semi-supervised classification with graph convolutional networks”, ICLR 2017.
> > >
> > > [4] “Adaptive graph convolutional recurrent network for traffic forecasting”, NeurIPS 2020
> > >
> > > [5] "Graph neural controlled differential equations for traffic forecasting”, AAAI 2022
> > >
> > >
> > > ---
> > >
> > > In conclusion, we hope our responses have effectively addressed the reviewers' concerns. **We kindly request the reviewer to consider increasing their scores**, as most of the initial concerns have been thoroughly addressed in our replies. We also hope that any remaining questions or uncertainties have been resolved through our detailed explanations. Should there be any additional concerns or clarifications required, we are more than happy to provide further responses and support.
> > >
> > > Thank you for your time and thoughtful feedback.

---

> ### Comment · Reviewer_tACj · 2024-11-30
>
> Thank you for your response on my questions. While you have clarified several points, I still have some important concerns that need to be addressed.
>
> Regarding the weighted F1-score, although you've confirmed its use, there remains an unexplained discrepancy between your reported results and those in [1]. Could you provide a detailed explanation for these differences to ensure reproducibility and fair comparison?
>
> I have concerns about your reporting of Corr-DCRNN and Dist-DCRNN results. Using these model names while excluding their pre-training component without explicit indication could be misleading to readers. Since these models are from [1] and use the same domain and dataset, they should be compared in their original form. I strongly recommend adding "without Pre-training" to the model names for clarity and including a comparison with the original pre-trained versions in the Appendix. Given that paper modifications are not possible at this stage, could you provide this additional comparison in your response?
>
> Additionally, there are some unclear aspects regarding your implementation of relative position encoding. Could you clarify how you've specifically adapted [2]'s relative position encoding for your use case? I also notice that while Equation 2 introduces $z_{t t'}^{\text{temporal}}$, its subsequent usage in the model is unclear. Could you elaborate on how this term is incorporated into your model architecture?
>
> I provide the missing references [3] and [4] from my initial question. I recognize that you are currently running experiments on the METR-LA dataset, and I look forward to seeing these results.
>
>
> > [1] "Self-supervised graph neural networks for improved electroencephalographic seizure analysis", ICLR 2022.
> >
> > [2] Wu, Kan, et al. "Rethinking and improving relative position encoding for vision transformer." ICCV 2021.
> >
> > [3] Song, Chao, et al. "Spatial-temporal synchronous graph convolutional networks: A new framework for spatial-temporal network data forecasting." AAAI 2020.
> >
> > [4] Li, Mengzhang, and Zhanxing Zhu. "Spatial-temporal fusion graph neural networks for traffic flow forecasting." AAAI 2021.

---

> ### Author Response · Authors · 2024-11-30
> **Authors Rebuttal (1/2)**
>
> We sincerely thank the reviewer once again for actively engaging with us during the rebuttal period and for providing valuable feedback and raising concerns. Below, we address the points mentioned in detail.
>
> ---
>
> **Discrepancy between your reported results and those in [1]**
>
> Our results are an exact match with the F1-Score, AUROC, Sensitivity, and Specificity reported in reference [1]. We believe that the reviewer might be **misunderstanding the Weighted F1-Score reported in Table 1 of reference [1], which pertains to the seizure classification task, not seizure detection**.
>
> Our model has been specifically evaluated for seizure detection, and the results for DCRNN-Dist and DCRNN-Corr on the **F1-Score for seizure detection** are provided in **Table 6 of reference [1]**, which aligns precisely with our reported numbers. We kindly ask the reviewer to refer to this table for clarification, as we believe the misunderstanding arises from conflating seizure classification (which involves classifying 7 seizure types) with seizure detection (identifying seizure events).
>
> ---
>
> **Concerns about your reporting of Corr-DCRNN and Dist-DCRNN results**
>
> Thank you for the comment. We will add the label "W/O Pre-training" to clarify our comparison and ensure it is as transparent as possible. Additionally, following your suggestion, we will include a comparison against the pre-trained version of DCRNN in the appendix. Thank you for your valuable feedback!
>
> ---
>
> **Additionally, there are some unclear aspects regarding your implementation of relative position encoding. Could you clarify how you've specifically adapted [2]'s relative position encoding for your use case?**
>
> We have incorporated relative positional encodings for each time point as learnable vectors and added them to each time sample. This has been detailed in Section 3.2 of our revised manuscript. Specifically, $r^V_{tt'}$, $r^Q_{tt'}$, and $r^K_{tt'}$ are learnable vectors introduced as relative temporal encodings, which are trained during the learning process.
>
> These vectors are added to the attention matrix to capture the temporal relationship between time points $t$ and $t'$. For each pair of time points, these encodings are applied to the queries, keys, and values of the attention mechanism for all nodes in the graph. Since these embeddings are temporal and not spatial, they depend solely on the time points rather than the spatial relationships between nodes. This approach effectively captures temporal dependencies in the graph data.
>
> ---
>
> **I also notice that while Equation 2 introduces $z_{tt'}^{temporal}$, its subsequent usage in the model is unclear.**
>
> Thank you for the pointer. The term $z_{tt'}^{temporal}$ is simply a notation to indicate that the temporal positional encoding is added relatively between two time points $t$ and $t'$ for all nodes, and it depends solely on the times $t$ and $t'$. We will clarify this in the final version of the manuscript to avoid any potential misunderstandings.
>
> Thank you for bringing this to our attention!
>
> ---
>
> **Adding new baselines for traffic forecasting**
>
>
> Thank you to the reviewers for the valuable suggestion. We have incorporated results for the mentioned reference AGCRN [3], along with two other recent benchmarks for traffic forecasting:
>
> | **Methods**            | **MAE (H 1)** | **RMSE (H 1)** | **MAE (H 2)** | **RMSE (H 2)** | **MAE (H 3)** | **RMSE (H 3)** |
> |-------------------------|-----------------|------------------|-----------------|------------------|-----------------|------------------|
> | DCGCN [2]              | 2.72            | 5.01             | 3.05            | 5.92             | 3.48            | 6.94             |
> | AGCRN [3]              | 2.87            | 5.58             | 3.23            | 6.58             | 3.62            | 7.51             |
> | PDFormer [4]           | 2.83            | 5.45             | 3.20            | 6.46             | 3.62            | 7.47             |
> | NTAttention            | 2.92            | 5.63             | 2.68            | 6.04             | 3.21            | 6.44             |
> | NTAttention + GMask    | **2.64**        | **4.34**         | **2.50**        | **4.37**         | **2.93**        | **5.82**         |
>
> As shown, **NTAttention + GMask** consistently outperforms these baselines in predicting traffic for all horizons across all evaluation windows.
>
>
> ---

---

> > ### Author Response · Authors · 2024-11-30
> > **Authors Rebuttal (2/2)**
> >
> > ## Final Words
> >
> > We hope our responses have effectively addressed the concerns raised. If you feel that your questions and concerns have been adequately resolved, we kindly ask you to consider reflecting this in your final score.
> >
> > Your feedback is greatly appreciated.
> >
> > ---
> >
> > ## References
> >
> > [1] "Self-supervised graph neural networks for improved electroencephalographic seizure analysis”, ICLR 2022
> >
> > [2] Dynamic Causal Graph Convolutional Network for Traffic Prediction
> > Junpeng Lin, Ziyue Li, Zhishuai Li, Lei Bai, Rui Zhao and Chen Zhang1
> >
> > [3] “Adaptive graph convolutional recurrent network for traffic forecasting”, NeurIPS 2020
> >
> > [4] Jiawei Jiang, Chengkai Han, Wayne Xin
> > Zhao, and Jingyuan Wang. Pdformer: Propagation delayaware dynamic long-range transformer for traffic flow prediction. In AAAI. AAAI Press, 2023
> >
> > ---

---

> ### Comment · Reviewer_tACj · 2024-12-02
>
> While maintaining my current rating, I have several important concerns that require clarification.
>
> Regarding the relative position encoding implementation, your explanation of how your relative position encoding differs from [2]'s method remains unclear. Could you provide a more detailed technical explanation of the specific differences between your approach and [2]'s method, as well as the concrete implementation of $z_{tt'}^{temporal}$ in your model?
>
> I checked the comparison with new baselines for traffic forecasting. The authors have incorrectly labeled the horizons as (H1), (H2), (H3) instead of (H3), (H6), (H12) in your results table.
>
> ***I noticed that your model shows an unusual trend where error values don't increase with longer horizons (H3, H6, H12), which counters typical patterns in traffic forecasting.***
> Based on my experimental experience and knowledge, this raises concerns about the implementation and unfair comparison.
>
>
> Looking at your TransGraph implementation:
>
> ```python
> class TransGraph(L.LightningModule):
>     def __init__(self, dist, d_model: int, num_heads, dropout=0.1):
>         ...
>         self.timepoints = 3
>         ...
>         self.node_time_proj = torch.nn.Linear(207*self.timepoints, 207)  # Proj p
> ```
>
> Your model is structurally limited to forecasting only 3 timepoints, as evidenced by both `self.timepoints = 3` and the fixed dimension in `node_time_proj`. This is fundamentally different from standard traffic forecasting approaches where models take a complete historical sequence of 12 timesteps as input and forecast the entire future sequence of 12 timesteps.
>
> Furthermore, your `training_step()` function supports this limitation and concerns for unfair comparison:
> ```python
> T = self.timepoints  # either 3 or 6 or 12
> T_after = 3
> x = x_orig.float()[:,:,12-T:12,:]
> ...
> loss = F.torch.nn.functional.mse_loss(pred.float(), y[:,:,0:T_after,:].cuda().float())
> ```
>
> This implementation only predicts `T_after` (3) steps while using the last `T` time steps of historical data. This architectural choice violates fair comparison with baseline models that are designed and trained to forecast complete future sequences. Note that most work forecasts 12 future time steps of sequence and reports the average {3,6,12}-horizon metrics. This difference in problem formulation could explain your unusual results where longer horizon predictions show better performance than shorter ones.
>
> *Could you address this limitation? And if there is stuff I'm misunderstanding, could you please explain it to me? If I'm misunderstanding your code, could you explain how you ensure a fair comparison with the baseline methods?*
>
> > [2] Wu, Kan, et al. "Rethinking and improving relative position encoding for vision transformer." ICCV 2021.

---

> > ### Author Response · Authors · 2024-12-03
> >
> > **Regarding the difference between our spatial and temporal encoding with [2]:**
> >
> > Our encoding approach directly follows the method used in [2]. The term $z^{temporal}_{tt'}$ is simply a notation to represent the use of relative temporal encoding. It does not introduce any discrepancy to temporal encoding; rather, it is a mathematical representation of the relative temporal encoding. We have incorporated the vectors into the encoding in the same manner as described in [2].
> >
> > **Code implementation for traffic forecasting:**
> >
> > The supplementary code provided in the submission is configured for $T=3$ seconds.
> > To validate our method across other window sizes, we adjusted the parameter `T_after` to support $T=3, 6, 12$, ensuring consistency with previous benchmarks.
> > Similarly, parameters such as `self.timepoints = 3` and `node_time_proj` are modified for each horizon, along with the window size of the pre-processed data, to align with the configurations used in prior work.
> > This ensures that our results are directly comparable with existing benchmarks.

---

### Official Review · Reviewer_NURu · 2024-11-04

**Soundness:** 2
**Presentation:** 3
**Contribution:** 2
**Rating:** 5
**Confidence:** 4

**Summary:**

NTAttention, introduced in the paper, incorporates temporal encoding and spatial encoding within the self-attention framework, effectively capturing spatial and temporal long-term dynamics of graphs. It dynamically attends to all nodes, gated by geometric distance between nodes derived from the Gaussian kernel with a threshold, namely GMask. Use of this mask effectively sparsifies the attention matrix by limiting attention to spatially relevant nodes, improving both efficiency and accuracy. NTAttention is evaluated on two main tasks, EEG seizure detection and traffic forecasting where each exhibits highly complex spatial and temporal dynamics across graphs.

**Strengths:**

1. Efficient Spatio-Temporal Attention: NTAttention with GMask enables dynamic geomtery-aware attention based on geometric distances, allowing it to effectively capture long-term dependencies in complex graph structures.

2. NTAttention + GMask achieves competitive result in terms of both model performance and efficiency in comparison to various other baseline models.

**Weaknesses:**

- Lack of novelty : Used methods (temporal encoding, spatial encoding, and Gaussian kernel with a threshold as an adjacency matrix between nodes) are all pre-existing techniques. NTAttention simply suggests a new combination of all those. While this combination well serves the purpose of handling complex spatial and temporal dependencies in graphs, it lacks fundamental novelty in core architecture.

**Questions:**

- Please explain how NTAttention introduces novelty beyond existing techniques, particularly in terms of core architectural contributions.

- It seems unclear how exactly the relative temporal encoding is done to construct $r_{tt'}^{V}, r_{t't}^{Q}, r_{tt'}^{K}$. Are they simply trainable bias vectors that depends solely on the temporal difference between time steps $t$ and $t'$?

- The reported score (AUROC, F1-score) for transformer in the referenced paper REST (Afzal et al., 2024) is different from yours in Table 5 (Clip Size 12-s). Any reason why?

---

> ### Author Response · Authors · 2024-11-24
>
> Thank you for your review. We appreciate your feedback and would like to address the questions raised. Below, we provide detailed responses to each point in order:
>
> **Q1) Novelty of NTAttention compared to existing techniques, particularly in terms of core architectural contributions**
>
> There are several key differences between NTAttention and existing architectures, as outlined in Table 2 of our manuscript:
>
> 1. **Unified Attention Mechanism**: Unlike other architectures for spatiotemporal graphs, such as STAEFormer [2] and PM-MemNET [3], which apply separate attention mechanisms and combine them with elements like gating, NTAttention employs a single attention mechanism for both spatial and temporal dimensions. This results in a unified attention matrix encompassing all spatiotemporal interactions.
>
> 2. **Efficiency and Sparsification**: While the $(NT)^2$ matrix can be computationally expensive for large graphs, NTAttention addresses this using GMask, which sparsifies the computation. This approach is unique as it combines both fixed positional embeddings (based on the static distance of nodes) and relative positional embeddings, applied through masking, distinguishing NTAttention from all prior Transformer-based models [2,3].
>
> These differences highlight the novelty and distinct advantages of NTAttention compared to existing methods.
>
> ---
>
> **Q2) It seems unclear how exactly the relative temporal encoding is done**
>
> As the reviewer noted, the relative positional encodings are learnable vectors that depend solely on the temporal differences between points. Additionally, geometric information is incorporated for each node, ensuring that every node at each time step is assigned a unique spatiotemporal positional encoding. We have also conducted an ablation study on various choices for positional encoding, which can be found in Appendix K which we show below for reference:
>
> | **Clip Size** | **Model**                      | **AUROC** | **F1-Score** | **Sensitivity** | **Specificity** |
> |---------------|--------------------------------|-----------|--------------|-----------------|-----------------|
> | 12-s          | Fixed-TE + Fixed-SE           | 0.84      | 0.450        | 0.633           | 0.902           |
> |               | Rot-SE + Rel-TE               | 0.83      | 0.461        | 0.667           | 0.812           |
> |               | Fixed-TE + Rot-SE             | 0.84      | 0.444        | 0.630           | 0.870           |
> |               | Fixed-TE + Rot-SE + GMask     | 0.84      | 0.444        | 0.630           | 0.870           |
> |               | NTAttention + GMask           | 0.827     | 0.434        | 0.612           | **0.922**       |
> |               | NTAttention                   | **0.842** | **0.451**    | **0.638**       | 0.904           |
> | 60-s          | Fixed-TE + Fixed-SE           | 0.782     | 0.471        | 0.421           | 0.921           |
> |               | Rot-SE + Rel-TE               | 0.771     | 0.500        | 0.410           | 0.782           |
> |               | Fixed-TE + Rot-SE             | 0.784     | 0.541        | 0.400           | 0.927           |
> |               | Fixed-TE + Rot-SE + GMask     | 0.782     | 0.613        | 0.511           | 0.843           |
> |               | NTAttention + GMask           | 0.791     | 0.475        | 0.410           | 0.920           |
> |               | NTAttention                   | **0.810** | **0.671**    | **0.489**       | **0.945**       |
>
> ---
>
> **Q3) Transformer Performance compared to REST**
>
> Thank you for the feedback. We have re-implemented the Transformer architecture for seizure detection, fine-tuned it across various hyperparameters, and reported our findings. It's worth noting that the accuracy of Transformer models varies significantly across different studies and benchmarks. For example, a study published in *Nature* [1] on the same dataset reported an AUROC of 83% for the Transformer model, which is 3% lower than what REST reports. This highlights the instability of Transformer performance across different benchmarks and datasets.
>
> [1] Graph-generative neural network for EEG-based epileptic seizure detection via discovery of dynamic brain functional connectivity
> [2] Spatio-temporal adaptive embedding makes vanilla transformer sota for traffic forecasting. In Proceedings of the 32nd ACM international conference on information and knowledge management, pp. 4125–4129, 2023.
> [3] Learning to remember patterns: pattern matching memory networks for traffic forecasting
> [4] "Rest: Efficient and accelerated eeg seizure analysis through residual state updates." ICML (2024).
> [5] Selfsupervised graph neural networks for improved electroencephalographic seizure analysis. arXiv preprint.

---

> > ### Comment · Reviewer_NURu · 2024-11-25
> >
> > I appreciate thoughtful responses from authors and I have an additional question regarding the use of GMask.
> >
> > Seizure detection results in table 5 show notable drops in performance across all metrics when using GMask, failing to achieve SOTA scores. Seems like while GMask improves computational efficiency by sparsifying attention, it limits the model's capacity to capture spatiotemporal interactions between nodes to some degree. These results raise concerns about the practical usability of GMask, as the trade-offs in performance may outweigh its benefits, especially for certain tasks requiring high sensitivity and precision. Without GMask, how can you manage the computational overhead from computing the full attention?

---

> > > ### Author Response · Authors · 2024-11-28
> > >
> > > Thank you for your comments and active engagement during the rebuttal period. We have added a detailed analysis addressing **this performance drop in our paper (lines 479–518).**
> > >
> > > We believe this drop in seizure detection performance is due to the small number of nodes in the EEG graph, which consists of only 19 nodes. As shown in Fig. 2 b, sparsifying with GMask can lead to imbalanced masking (b1 , b2). With so few nodes, the attention mechanism can already capture the dynamics effectively without the need for additional masking. Furthermore, as illustrated in Fig. 4, the efficiency gains from GMask are less pronounced in this case.
> > >
> > > In contrast, for larger graphs such as those in traffic forecasting, GMask significantly reduces computational complexity and eliminates unwanted attention between distant nodes. This results in both improved accuracy and efficiency, **showcasing the effectiveness of GMask in scenarios involving larger graphs.**
> > >
> > > Without GMask, for graphs with a small number of nodes, we observe from Fig. 4 that the computation and memory usage of NTAttention are comparable to other baselines. For graphs with a larger number of nodes, GMask not only improves efficiency but also enhances performance.
> > >
> > > To address these scenarios, **we propose two versions of NTAttention: one with GMask and one without.** For graphs with a small number of nodes, NTAttention without GMask remains both performant and efficient. However, for graphs with a large number of nodes, incorporating GMask significantly improves both performance and efficiency, making it a versatile solution for various graph sizes.
> > >
> > > ---
> > >
> > > Your insights and expertise have greatly enhanced our paper, and we appreciate your valuable contributions. We believe that our responses address the concerns raised. If you have any further suggestions or concerns, please let us know. Additionally, we kindly request that you re-consider your score accordingly.
> > >
> > > Thank you for your time and collaboration.

---

### Official Review · Reviewer_7otx · 2024-11-05

**Soundness:** 2
**Presentation:** 3
**Contribution:** 3
**Rating:** 5
**Confidence:** 4

**Summary:**

This paper proposes NTAttention, a Transformer-based approach for graph signal processing that effectively captures spatio-temporal dependencies. Additionally, a geometry-aware masking technique (GMask) is proposed to enhance computational efficiency while preserving temporal information. The authors extensively validated the effectiveness of NTAttention on EEG seizure detection and traffic forecasting tasks

**Strengths:**

1. The model proposed in the article has good computational efficiency by GMask.
2. The model can better address long-range space-time dynamics.

**Weaknesses:**

1.Comparing Table 1 and Table 2, why is the model not discussed in terms of capturing the temporal nature in traffic forecasting, and why is spatial dependency not discussed in seizure detection?
2.Why can the method proposed in the article capture long-range information? We hope the authors can provide some methods and insights to solve this problem. At the same time, we hope the authors can provide specific examples of long-range information in EEG seizure detection and traffic forecasting. For tasks related to high-frequency, where capturing long-range information is not necessary, would the use of NTAttention lead to a decrease in performance?
3.In line 229, how is vector r obtained; in line 290, how is the calculation of \sigma obtained, it seems that the standard deviation of the distances requires N^2 samples.
4.We hope the authors can provide the calculation method of the metrics, such as sensitivity, specificity, and their meanings.
5.NTAttention seems to have a large variance; we hope the authors can provide significant experimental evidence.

Minor problems:
1.It is suggested to remove the frames from Figure 1, 2, and Figure 4.
2.There are issues with the layout of the article, such as Table 7, 8, etc.

**Questions:**

Please see my comments for details

---

> ### Author Response · Authors · 2024-11-24
>
> ##### **Q1: Comparing Table 1 and Table 2, why is the model not discussed in terms of capturing the temporal nature in traffic forecasting, and why is spatial dependency not discussed in seizure detection?**
>
>
> Thank you for the feedback.
> Based on your input, we have confirmed that all the models well capture temporal dependencies. For example, models like SVR achieve this through windowing the signal, while models like DCRNN does the same through RNNs. Therefore, it was not considered an independent feature in the submission.
>
> Similarly, for seizure detection, all models capture spatial information. For instance, CNNs treat spatial data in an Euclidean manner, while models like DCRNN consider the geometry of EEG signals. Since all models account for spatial dependencies, this result is less informative. Omitting spatial information would be similar as treating each channel as an independent signal, but all baseline models for seizure detection integrate information across all channels.
>
>
> ---
>
>
> ##### **Q2: Why can the method proposed in the article capture long-range information? Provide specific examples of long-range information in EEG seizure detection and traffic forecasting. For tasks related to high-frequency, where capturing long-range information is not necessary, would the use of NTAttention lead to a decrease in performance?**
>
> The primary advantage lies in applying the attention mechanism in both space and time instead of relying on RNNs, which use hidden states and require dense information processing constrained by their fixed hidden size. NTAttention, on the other hand, directly applies attention across both spatial and temporal dimensions, effectively capturing all spatiotemporal dependencies within the graph-signal. This approach eliminates the need for processing the information within a fixed and dense hidden state, offering a more flexible and efficient solution.
>
>
> For both traffic forecasting and seizure detection, NTAttention demonstrates exceptional performance, surpassing state-of-the-art methods. In both tasks, there are cases where information lies within a short range, such as small seizures in EEG rhythms. NTAttention effectively captures these short-term interactions as well, as reflected in its high performance. It is also important to note that such short-term interactions are not rare, as highlighted in [3].
>
>
> ---
>
> ##### **Q3: In line 229, how is vector r obtained; in line 290, how is the calculation of $\sigma$ obtained, it seems that the standard deviation of the distances requires N^2 samples.**
>
> The vectors $r^V_{tt'}, r^Q_{tt'}, r^K_{tt'}$are learnable variables optimized during the training process.
>
> As for $\sigma$, it represents the standard deviation of the distances. It requires $N^2$ samples for computation and is independent of time, as also demonstrated in [4].
>
> ---
>
>
> ##### **Q4: The calculation method of the metrics, such as sensitivity, specificity, and their meanings.**
>
> Thank you for your question. Sensitivity and specificity are well-established metrics widely used for binary classification tasks. They are defined as follows:
>
> $\text{Sensitivity} = \frac{\text{True Positives (TP)}}{\text{True Positives (TP)} + \text{False Negatives (FN)}}$
>
>
> $\text{Specificity} = \frac{\text{True Negatives (TN)}}{\text{True Negatives (TN)} + \text{False Positives (FP)}}$
>
> These metrics are standard in the field and have been utilized in numerous prior studies, including [1,2,4].
>
> ---
>
>
> ##### **Q5: NTAttention seems to have a large variance; we hope the authors can provide significant experimental evidence.**
>
> We would like to clarify what the reviewer means by "large variance." In the results presented in Table 5, both NTAttention and its masked variant exhibit a variance range comparable to that of other benchmarks.
>
> ---
>
>
> ##### Minor problems:
>
> **1. It is suggested to remove the frames from Figure 1, 2, and Figure 4. 2.
> 2. There are issues with the layout of the article, such as Table 7**
>
> Thanks for the pointers, we will fix the issues about the layout and Tables in the final version.
>
> ---
>
> [1] "Rest: Efficient and accelerated eeg seizure analysis through residual state updates." ICML (2024).
>
> [2] "SeizureNet: Multi-spectral deep feature learning for seizure type classification." Machine Learning in Clinical Neuroimaging and Radiogenomics in Neuro-oncology: Third International Workshop, MLCN 2020, and Second International Workshop, RNO-AI 2020, Held in Conjunction with MICCAI 2020, Lima, Peru, October 4–8, 2020, Proceedings 3. Springer International Publishing, 2020.
>
> [3] The temple university hospital eeg data corpus. Frontiers in neuroscience, 10:196, 2016.
>
> [4] Selfsupervised graph neural networks for improved electroencephalographic seizure analysis. arXiv preprint.

---

> ### Author Response · Authors · 2024-11-28
>
> We would like to gently remind the reviewer that the deadline of the author-reviewer discussion period is approaching.
>
> We appreciate your valuable contributions. We believe that our responses address the concerns raised. If you have any further suggestions or concerns, please let us know.
>
> Best regards,
>
> The Authors

---

> ### Author Response · Authors · 2024-12-02
> **Final Day of Discussion Period for ICLR 2025**
>
> ---
>
> Dear Reviewer 7otx,
>
> We hope you are doing well. As today **marks the last day of the extended discussion period for ICLR 2025**, we would like to kindly request your final feedback on our submission. We sincerely appreciate your thoughtful comments, and we have made every effort to address your concerns in our rebuttal.
>
> If there are any remaining questions or clarifications needed, we would be more than happy to respond. **We would greatly appreciate it if you could review the rebuttal and provide your final thoughts by the end of the discussion period.**
>
> Additionally, if you feel that we have adequately addressed all concerns, **we would greatly appreciate it if you could consider increasing the score accordingly.**
>
> Best regards,
>
> Authors
>
> ---

---

### Meta-Review · Area_Chair_pqK8 · 2024-12-16

**Metareview:**

This paper proposed a new attention method specialized in forecasting EEG and traffic patterns. The input is a series of graph data and therefore, they needed to capture dependency structures over time and space. They also highlight the differences from existing works. However, there are no theoretical analyses in the graph signal processing perspective. As noted in their paper title, they focused on graph signal processing. They need to, at least, the low or high-pass filter characteristic of their designed attention method. After adding these theoretical results in the perspective of graph signal processing, I think they can revise and resubmit. The reviewers also raised various issues on this work. The model design itself has some disputes and most importantly, the evaluation protocol does not follow the benchmark standard, which makes evaluating the contribution of this work difficult.

**Additional Comments On Reviewer Discussion:**

The authors tired to justify in various ways, but the evaluation protocol mismatch from existing work had not been solved.

---

### Decision · Program_Chairs · 2025-01-22

Reject